# Research on Object Detection and Recognition Method for UAV Aerial Images Based on Improved YOLOv5

**Heng Zhang** , **Faming Shao \***, **Xiaohui He** , **Zihan Zhang, Yonggen Cai and Shaohua Bi**

College of Field Engineering, Army Engineering University of PLA, Nanjing 210007, China;
zhangheng4216@sina.com (H.Z.); gcbhxh@aeu.edu.cn (X.H.); zhzhang1126@163.com (Z.Z.);
caiyonggen@163.com (Y.C.); 17337434454@163.com (S.B.)
**\*** Correspondence: shaofaming@126.com; Tel.: +86-185-4985-4591

**Abstract:** In this paper, an object detection and recognition method based on improved YOLOv5 is proposed for application on unmanned aerial vehicle (UAV) aerial images. Firstly, we improved the traditional Gabor function to obtain Gabor convolutional kernels with better edge enhancement properties. We used eight Gabor convolutional kernels to enhance the object edges from eight directions, and the enhanced image has obvious edge features, thus providing the best object area for subsequent deep feature extraction work. Secondly, we added a coordinate attention (CA) mechanism to the backbone of YOLOv5. The plug-and-play lightweight CA mechanism considers information of both the spatial location and channel of features and can accurately capture the long-range dependencies of positions. CA is like the eyes of YOLOv5, making it easier for the network to find the region of interest (ROI). Once again, we replaced the Path Aggregation Network (PANet) with a Bidirectional Feature Pyramid Network (BiFPN) at the neck of YOLOv5. BiFPN performs weighting operations on different input feature layers, which helps to balance the contribution of each layer. In addition, BiFPN adds horizontally connected feature branches across nodes on a bidirectional feature fusion structure to fuse more in-depth feature information. Finally, we trained the overall improved YOLOv5 model on our integrated dataset LSDUVD and compared it with other models on multiple datasets. The results show that our method has the best convergence effect and mAP value, which demonstrates that our method has unique advantages in processing detection tasks of UAV aerial images.

**Keywords:** UAV aerial images; YOLOv5; Gabor; edge enhancement; coordinate attention mechanism; bidirectional feature pyramid network

## 1. Introduction

In recent years, drones have been used in a large number of industries due to their advantages of being lightweight and convenient to use. In the field of artificial intelligence, people use convolutional neural networks based on deep learning to recognize and analyze objects in aerial images [1]. The drone aerial images are very rich in information, which is of great value in intelligence reconnaissance, geographic surveying, and research. Therefore, drone aerial images are the main object of our research.

When unmanned aerial vehicles are conducting aerial photography missions, image quality may be degraded by various interferences, such as the environment around drones, body vibrations, and equipment performance. These interference factors can cause blurring, unclear object texture, noise, and other issues in the images collected by drones. In addition, as the height of the drone increases, the image coverage of the scene continues to increase, and the changes in object scale gradually become apparent, which leads to the weakening of object features. These problems make object detection in drone aerial images difficult, so much research has been conducted on detection algorithms for drone aerial images. Chen Y et al. [2] utilized the MobileNetv3 model and improved the backbone structure in YOLOv5

to solve the problem of excessive memory usage when detecting high-resolution images of drones, thereby improving network memory utilization. Messmer M et al. [3] proposed a height-adaptive image preprocessing method that can adjust the size of objects in the drone images to meet model requirements. Experimental results have shown that this method improves the detection accuracy and speed of objects in the aerial view of drones and can be adapted to all state-of-the-art detectors. Due to the possibility of abnormal objects invading railways and affecting the safety of railway traffic, unmanned aerial vehicles are often used to monitor such targets. Therefore, in response to the issue of real-time drone detection of railway foreign objects, Yundong L et al. [4] proposed a deep learning-based multi-block SSD (Single Shot Multi-Box Detector) detection method, in which the original image is divided into several sub-images and then transported to the multi-block SSD network. The purpose of this is to increase local contextual information and enable the SSD network to better utilize real-time and high-precision capabilities. Cai H et al. [5] proposed a lightweight detection method based on YOLOv4. They changed the DarkNet-53 network of YOLOv4 to a MobileNet network, improved the neck feature extraction network and head prediction network, and ensured good real-time performance while greatly reducing parameters. This is very suitable for the lightweight characteristics of unmanned aerial vehicles. The high density and overlap of small objects in the shadow of photovoltaic panels pose great difficulties for real-time detection. Jun W et al. [6] improved the RetinaNet algorithm model. Firstly, they proposed the Ghost CSP DenseNet feature extraction network to reduce network size and improve network detection speed. Secondly, the model uses the Ghost model and recursive feature fusion mechanism for feature fusion and adjusts the feature layer to adapt to multi-scale targets, which can improve feature expression ability and detection speed. Finally, the model uses the SiLU activation function to improve the network learning ability. In addition, the network uses the CioU regression loss function to improve the network prediction ability and convergence speed. The experimental results show that the model has better mAP, with a model size of only 8.75 MB and a detection speed of up to 50.7 FPS. These studies indicate that object detection algorithms currently do not have universality across different application fields. Therefore, adjusting algorithms appropriately for different application fields is a necessary measure before AI algorithms become stronger.

In this article, we mainly solve the background problem, object distribution problem, and object scale problem of UAV aerial images from two aspects. On the one hand, the image is preprocessed. We use an edge processing method based on the edge features of the object to filter and strengthen the edges in multiple directions. The strengthened object not only has obvious edge information but can also effectively filter out redundant background information. We attach the processed image to the original image, highlighting the object area and thus making it easier for the detection model to find the target during feature extraction. It is then fed into the detection model of YOLOv5 along with the original image. On the other hand, this article introduces an attention mechanism and a feature pyramid structure that enhances weights. The addition of an attention mechanism allows the model to simultaneously consider channel and spatial information and has high recognition and localization efficiency in detecting multi-scale objects, especially small objects with a dense arrangement, overlap, and shadow occlusion. The feature pyramid structure with enhanced weights is used to fuse the feature layers input from the YOLOv5 backbone in multiple ways and perform weighting operations based on the contribution of the input feature layers. This enables the feature layer to have rich semantic information and location information after multiple fusion events. In order to improve the generalization ability and robustness of the model proposed in this article, we integrate several datasets and conduct comparative experiments and precision–recall statistics on the overall improved model and other models. The experimental results verify the efficient detection ability of the model proposed in this article.

In this article, our main contributions are as follows:

- Gabor filter banks are used to preprocess the edges of objects. We improve the traditional inefficient Gabor function by using discrete quantization. Due to the fact that a single Gabor convolution kernel can only enhance the edges of objects in a single direction, we utilize multiple improved Gabor convolution kernels (filters) to enhance the edges of objects from different directions. From the enhanced image, it can be seen that the redundant background is suppressed and the edge features of the object are obvious.
- CA is added to the backbone of YOLOv5. CA embeds spatial information of object features into channel information to reduce information loss and enable the network to accurately capture the long-range dependencies of positions. The introduction of CA is beneficial for the model in better identifying target areas, with good effectiveness in locating small targets.
- PANet is replaced by BiFPN on the neck of YOLOv5. Developed on the basis of PANet, in BiFPN (1) the network is simplified by deleting nodes without feature fusion and with little contribution to output, (2) an additional feature branch is added across intermediate nodes between the original input and output nodes in the same layer to fuse feature information from more layers, and (3) input feature layers with different resolutions are weighted to balance their respective contributions, which is beneficial for improving training speed and efficiency.
- The new dataset LSDUVD (Large-Scale Dataset Based on UCAS-AOD, VisDrone2019, and DOTA-V1.0) is obtained by integrating UCAS-AOD, VisDrone2019, and DOTA-V1.0 through data augmentation methods such as flipping, random cropping, and cutout.

The organizational structure of this article is as follows. Section 2 discusses related studies on object detection and image processing. Section 3 introduces the overall working structure of our method and the details of each improvement module. Section 4 mainly introduces our dataset, model experiments, and analysis of the results. Section 5 is the summary and outlook for the article.

## 2. Related Works

### 2.1. Object Detection

Object detection is a very important technology in the field of artificial intelligence computer vision. In addition to being widely used in facial recognition [7], intelligent transportation [8], industrial detection [9], and environmental monitoring [10], it also plays an increasingly important role in the recognition of UAV aerial images [11]. Given the increasingly strong demand for technology in the development of an intelligent society, people's requirements for this technology are also increasing. Therefore, the upgrading and iteration of object detection technology are among the most challenging issues at present.

From the perspective of object detection algorithms, traditional object detection algorithms have many shortcomings in terms of detection accuracy and speed. One reason is that the computing resources at the time of their development were insufficient, and the other important reason is that they use a cumbersome sliding window method to traverse the entire image, find a certain number of candidate boxes, and then manually extract features. This results in the loss of much important information, and manual feature extraction has great limitations, such as that the feature robustness is not strong [12]. Common traditional object detection algorithms include the Viola Jones detector [13], the HOG detector [14], and the component-based deformable model (DPM) [15]. Of these, the Viola Jones detector is mainly composed of three parts: Harr features [16], Adaboost classifier [17], and cascaded classifier. Because Harr features are relatively simple features, the classifier is prone to overfitting, which leads to low robustness of the algorithm. The HOG detector combines HOG features [18] with SVM classifiers [19] and has good performance in pedestrian detection. However, its detection window does not change with the object scale, and this can only be achieved by repeatedly scaling the size of the input image. DPM requires the manual design of the corresponding incentive templates for different objects, which has a significant effect on a small number of objects. However, this approach is

clearly not suitable for multi-type and multi-scale objects. Since 2014, breakthroughs have been made in deep learning research. Convolutional neural network algorithms based on deep learning have been widely applied in object detection, and object detection technology has also achieved unprecedented development. For example, in the study of autonomous landing missions for UAV, researchers in [20] improved the deep learning-based model TLD by combining the Hough Transform, enabling the model to effectively detect and track objects, helping the drone better land on moving ground vehicles. In order to improve the accuracy of pedestrian detection in drone aerial images, YaChing C et al. [21], based on YOLOv2, utilized image segmentation and vanishing point transformation preprocessing methods to overcome the difficulties caused by small objects and object deformation in aerial images.

The current algorithms are divided into two types: one-stage and two-stage algorithms. The difference between the two is that the one-stage algorithm does not generate candidate boxes, and the network directly classifies and regresses objects. By contrast, the two-stage algorithm first generates candidate boxes by the network and then performs object classification. Representative one-stage algorithms include SSD [22], YOLO [23], etc., while two-stage algorithms include R-CNN [24], Fast-RCNN [25], Faster-RCNN [26], SPPNet [27], etc. Of these, the one-stage algorithm model is lightweight and has an advantage in computational speed, making it very suitable for processing real-time tasks, but it has shortcomings in recognition accuracy. Although the classification and regression process of two-stage algorithms is relatively cumbersome, their detection accuracy is relatively high. YOLO, as a typical one-stage algorithm, has a simple structure consisting of convolutional layers, pooling layers, and fully connected layers. The input image is transformed by a neural network, and the tensor is directly output. YOLO divides the input image into grids, and each grid predicts the object categories in the bounding box and outputs the probability of the categories. The first-generation version of YOLO, YOLOv1 [28], was released in 2015 as the first installment of the YOLO series, laying the foundation for subsequent versions of YOLO. The subsequent versions are all improvements and innovations based on this foundation. YOLOv1 uses a network to regress objects and implement a unified system, but due to the limitations of its network size, it has limitations in space and scalability. At the end of 2021, YOLO was iterated to the fifth version. Of the various versions, YOLOv2 [29] uses Darknet-19 as the network subject, which reduces the computational complexity of the network by four convolutional layers compared to YOLOv1. In 2018, YOLOv3 [30] was published, and YOLOv3 has made significant improvements compared to previous versions. Its backbone network is Darknet-53, and this deeper convolutional layer combined with FPN architecture improves the detection performance of multi-scale objects. The YOLOv4 [31] model conducts more optimization work on the network, such as adding multiple CSP modules to the backbone, which can effectively improve the learning ability of the network. The neck and output head are partially optimized for feature extraction and loss function, respectively, which improves the detection accuracy of the network. In 2020, after the release of YOLOv4, YOLOv5 was released. Although the original authors of YOLO did not publish papers on this version, it can be seen from other related papers, such as [32,33], that YOLOv5 is a masterpiece of previous versions, with excellent performance in detection accuracy and speed.

*2.2. Image Preprocessing*

Before object detection in the image input model, researchers sometimes use image preprocessing to improve image quality, enabling the model to better recognize and locate objects. Zhou B et al. [34] proposed a multi-operator-based image edge processing scheme, which enhances the edge features of images through single-operator preprocessing and multi-operator serial parallel preprocessing. Combined with the SSD algorithm model, validation was conducted on an industrial dataset, and the results show that this method can effectively detect industrial defects. Chousangsntorn C et al. [35] used an image processing method based on OpenCV python [36] to detect defects in the serial number

printed on the hard drive block. In order to balance the contrast of the region where the serial number is located, they first used an image processing method based on OpenCV python to eliminate variations that cause changes in image contrast. Secondly, they used the Otsu approach [37] image preprocessing method to process the image threshold into binary. Furthermore, contour approximation technology was used to enhance the edges of the region of interest (ROI) and to achieve consistent image size through ROI cropping. In the second part, the author used four classification models, YOLOv4 and DarkNet-19 [38], EfficientNet-B0 [39], Deep Residual Network (ResNet-50) [40], and DenseNet-201 [41] to detect the sequence number of hard disk sliders. The detection accuracy exceeded 99%, and the shortest inference time was 256.91 ms. In the field of medical imaging, traditional X-ray imaging is an important reference for doctors to judge the focus, but for some diseases in the early stage of development, such as breast cancer, radiologists are difficult to find and make accurate predictions. To address this issue, Rehman K et al. [42] proposed a deep separable deep convolutional neural network. In this network, they used an automatic image preprocessing method to convert the original X-ray image DICOM format to PNG format and adjusted the image size by using the OpenCV resize image preprocessing method. These methods can effectively help the network classify lesions in the later stage. Gendall L et al. [43] proposed an object-based image analysis method for determining the survival of Canadian kelp. In this method, they preprocessed the remote sensing images of kelp. Firstly, they used ground control points and nearest neighbor interpolation to correct the geometric deformation of the images in ArcGIS. Secondly, geographic correction was performed on the overlapping images to restore them to the position before the overlap, and then root mean square error (RMSE) was used to determine the correction quality. Once again, all types of images were processed through reflectivity comparison, contrast adjustment, and adding pixel buffers. Finally, a visual evaluation was conducted on the combination of the highest-scoring band index or ratio with the visible band to select the best classification combination. When detecting vehicles in aerial images, Ref. [44] proposed an image preprocessing method based on Principal Component Analysis (PCA). This method removes common misclassifications caused by the ambiguity of small objects by removing unnecessary features, improving the detection accuracy of small vehicles in aerial photographs. When PCA is combined with ResNet50, the misclassification rate of vehicles in aerial images is significantly reduced, providing a better detection rate compared to existing baselines. The images that artificial intelligence relies on are obtained through optical sensors, and these sensors are designed to cater to human visual characteristics, rather than specifically designed for artificial intelligence algorithms. In fact, different intelligent organisms, such as frogs, cats, flies, etc., have significant differences from the human eye regarding the images they perceive. We believe that this difference matches their biological intelligence processing systems. This also indicates that different algorithms and applications have more closely matching image expression strategies. Under the premise that the current image acquisition method cannot be changed, strengthening the key information of the image through preprocessing should be a more effective strategy.

### 2.3. Attention Mechanism

In order to enable the network to extract more accurate features of the object area, attention mechanisms are sometimes introduced into network models by researchers. From a large number of research results, it can be seen that attention mechanisms can not only help networks effectively extract features but also improve network detection efficiency and reduce network computing burden. Due to the low accuracy of the original YOLOv5 model in detecting small objects in drone aerial images, the SENet attention mechanism was introduced into the YOLOv5 model in [45] to address this issue. Different channels of the feature map were weighted through squeezing and excitation, and the channel information of the feature map was strengthened, which is beneficial for the network to extract feature information. In order to reduce model parameters and save computational resources, Zhang B et al. [46] introduced a channel attention mechanism

in the improved feature extraction module ISFCREM. By weighting channels in different feature layers, the feature attention of different channels was improved, compensating for the loss of feature information caused by channel compression and enhancing the detection efficiency of the object area. Sometimes, influenced by the shooting equipment, the resolution of the object in remote sensing images may be small, and the image may be accompanied by noise interference, which brings great difficulties to the detection and recognition of targets. Huang M et al. [47] proposed a residual attention function fusion method to enhance the network's performance in detecting small objects. This method enhances the feature expression ability of multi-scale feature layers by integrating multi-scale contextual information. Based on this, they used a spatial attention mechanism (GPC) for convolutional compression of global pixels, which weights the global pixels of the original image. This can significantly reduce noise interference in the original image, highlight key pixel information in the image, and improve the network's ability to extract feature information. Dong M et al. [48] proposed a crawl detection model based on its own semantic information. In order to effectively capture the semantic feature information of objects in the real world, they designed a target feature attention mechanism based on the ROI method. This mechanism focuses on the feature information of the target through semantic information, separates the target feature information from the background information, and makes the target information unique. This enables the model to capture the target feature through semantic information. The experimental results of this method on the dataset have high accuracy. In a multiple-object tracking (MOT) task, the network may overlook the continuous motion information of the object and the attention of undifferentiated discovery and recognition, which greatly affects the tracking accuracy of the network toward the object. To address the above issues, Yifeng W et al. [49] proposed a Motion and Correlation-Multiple Object Tracking (MAC-MOT) method, in which a motion-enhanced attention module (MEA) and a dual correlation attention module (DCA) were introduced. MEA performs differentiation on adjacent feature layers, enhances motion correlation between feature layers, and suppresses irrelevant information between adjacent layers. The DCA module decouples and thereby separates the detection task and the recognition task, reducing conflicts and achieving a balance between the two tasks. After experimental evaluation, the proposed method can achieve ideal tracking and detection performance. The attention mechanism essentially involves using a certain processing strategy to reduce the randomness of the data in the image matrix, resulting in unequal data in different regions, making the image in the object region more effective in connecting the data processing channels. The essence of big data is to transform it into small data through a certain strategy, which requires the fusion of effective data and the early abandonment of some invalid data.

*2.4. Feature Fusion*

At present, research on feature fusion is becoming increasingly in-depth, and the methods and application forms of feature fusion are also becoming more diverse. It can be divided into early fusion and late fusion depending on the order of fusion and prediction. Early fusion refers to the fusion of multi-layer features first, followed by result prediction. Common early fusion methods include concat and add. Late fusion refers to the detection of partial fusion layers before the completion of all fusions, forming multiple detection results, and finally fusing multiple detection results. Common late fusion methods include SSD and Feature Pyramid Network (FPN). In underwater image object detection, a lightweight adaptive feature fusion network (LAFFNet) [50] is proposed to reduce the excessive storage occupied by the model and reduce model parameters. This network uses multiple Adaptive Feature Fusion (AFF) modules of different sizes to effectively fuse the generated multi-scale features with channel attention. Experiments have shown that this network reduces parameters by about 94% compared to GFLOP networks and has a more compact structure and lighter weight. Due to the influence of complex backgrounds or missing features on small objects in remote sensing images,

detection networks often cannot work effectively. In [51], a multiple-path feature pyramid network (MPFP-Net) is proposed. In the feature fusion section, the network uses an FPN architecture to connect feature layers of different scales, which is used to adjust the number of channels and the size of feature maps so that the feature layers maintain the gradient specification of the pyramid. FPN uses bottom-up and cross-connected networks that can fuse feature layers of different sizes, helping the network collect small object features and achieve more accurate detection accuracy. Guoyi Y et al. [52] proposed a feature fusion method called the multi-flow feature fusion module (MF3M) to address the issue of scale changes in object detection, building a door module and multiple information flows in MF3M to suppress redundant information in the feature mapping process and improve the integrity and accuracy of information transmission. They also proposed an improved deformable convolution algorithm, task-adaptive convolution (TaConv), which uses TaConv to output two feature maps on each layer of their constructed feature pyramid for classification and regression. This approach solves the internal contradiction between classification and regression, thereby improving model detection performance. Similarly, in order to solve the problems caused by object-scale changes in detection work, Ref. [53] proposed a feature fusion network (FFNet), which consists of two parts: a relationship fusion module (RFM) and a numerical fusion module (NFM). Long-range dependency information in RFM strengthens the information region of features and reduces interference from useless regions. In NFM, average operation is used to generate fusion weight values, and the corresponding region information is retained by determining the weight size such that information can be effectively transmitted in feature fusion. Through a large number of experiments, it was verified that the feature fusion strategy they designed exhibits an efficient detection performance for multi-scale targets, especially small objects. In the process of feature convolution from bottom to top, information such as position and texture will be lost layer by layer, but the semantic information of the feature layer will also be gradually enriched. Therefore, Ref. [54] proposed a feature fusion single-shot multi-box detector (FFSSD) based on a feature pyramid network (FPN), which adds multiple lateral connection structures on top of the FPN to improve the detector's feature extraction ability. Its up-and-down interaction mechanism fuses the high-level semantic information with the underlying location information, and the fused feature layer has more abundant deep information. This method has high mAP and strong real-time performance when applied to the task of prostate capsule detection.

## 3. Methods

### 3.1. Overview of the Proposed Method

The overall structure of the drone aerial image object detection model proposed in this article is shown in Figure 1. The overall structure is mainly composed of the data collection and enhancement section in the upper left region in the figure below, the image edge preprocessing section in the lower left region, and the improved YOLOv5 detection section in the right half region. The specific steps of the drone aerial image detection process are as follows. Firstly, to ensure the generalization ability of the model after training, we increased the data size of the three existing datasets through vertical flipping, horizontal flipping, random cropping, and cutout methods to obtain a new dataset named LSDUVD. Secondly, we used a combination of Gabor filters to enhance the image and then combine the filtered image with obvious edges after enhancement with the original image. Finally, we adjusted the resolution of the combined image to $640 \times 640$, and the image was then input into the improved YOLOv5 network. In the improved YOLOv5 model, the backbone network part that introduces a CA [55] mechanism will extract key region features from the input image. Then, in the neck network of the model, BiFPN [56] performs sampling and weighting operations on the input feature layer, which enables the full fusion of feature layers with different information. The improved YOLOv5 head performs object prediction on the fused features and outputs the results.

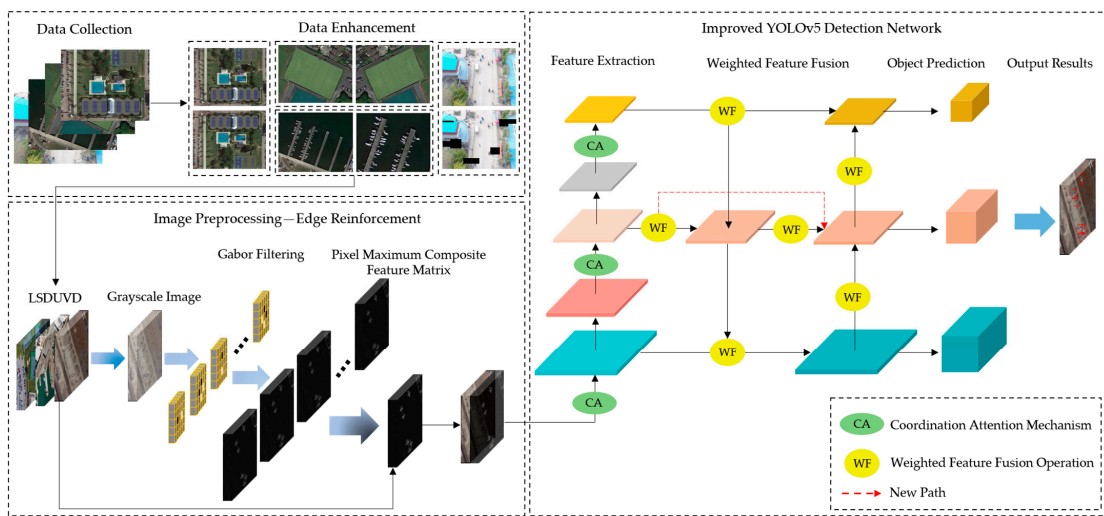

**Figure 1.** The overall structure diagram of the drone aerial image object detection model proposed in this article.

### 3.2. YOLOv5 Network Model

YOLOv5 is the fifth-generation version of the YOLO (You Only Look Once) series and is a single-stage object detection algorithm that belongs to the regression series of object detection methods. Unlike sliding window and subsequent region division detection methods, it treats the detection task as a regression problem, using a neural network to directly predict the coordinates of the bounding box, the confidence level of the objects in the bounding box, and the probability of the object's category from the entire image, allowing the achievement of end-to-end detection performance optimization. According to the different depths and widths of the network, YOLOv5 has four versions: YOLOv5s, YOLOv5m, YOLOv5l, and YOLOv5x. Among them, the YOLOv5s model has the smallest size and the fastest detection speed, which is widely used in the field of real-time object detection. So, the research work of this paper is based on YOLOv5s. This algorithm adds some new improvement ideas on the basis of YOLOv4, which greatly improves its speed and accuracy. For example, at the input end of the model, some improvement ideas have been proposed, mainly including mosaic data enhancement, adaptive anchor box calculation, adaptive image scaling, etc. In the model backbone network, some new ideas from other detection algorithms are integrated, mainly including the focus structure and CSP structure, which greatly reduces the number of network parameters. In the object detection network of the neck network, a PANet [57] structure is often inserted between the feature extraction backbone and the final head output layer. At the head output end, the anchor frame mechanism at the output end is the same as YOLOv4. The main improvements are the loss function GIOU-Loss during training and the DIOU-NMS filtered by the prediction frame. The structure of the YOLOv5s model is shown in Figure 2.

### 3.3. Improvement of YOLOv5 Model

Although a large number of cases have confirmed that the YOLOv5 model has good object detection performance, in the research on object detection in drone aerial images, due to the complex background, blurred edges, and small and dense targets of drone aerial images, YOLOv5 often cannot perform well when directly used for detection. Therefore, in this work, the YOLOv5 model is improved. At the input end of the model, images are preprocessed with edge enhancement. We use an improved Gabor filter bank to filter and calculate images, thereby strengthening the target edges from multiple directions, which can reduce the interference of complex background information and facilitate feature extraction. In the backbone network, we introduce a coordinate attention (CA) mechanism. On the premise that the target edge has been strengthened, CA can better integrate channel and position information, enhancing the network's positioning and recognition of the

target area. In the neck network, due to the fact that the original PANet structure does not distinguish different proportions of features during feature fusion, it simply adds them and directly outputs the feature map, resulting in low prediction accuracy of the model for small targets. Therefore, we introduce a bidirectional feature pyramid structure (BiFPN). BiFPN mainly balances the contribution of each feature map by adding weights to feature maps of different scales and adds branches of feature fusion across nodes, enabling the network to fuse more feature information, thereby improving the model's performance in object detection. The structure of the improved YOLOv5s object detection network is shown in Figure 3.

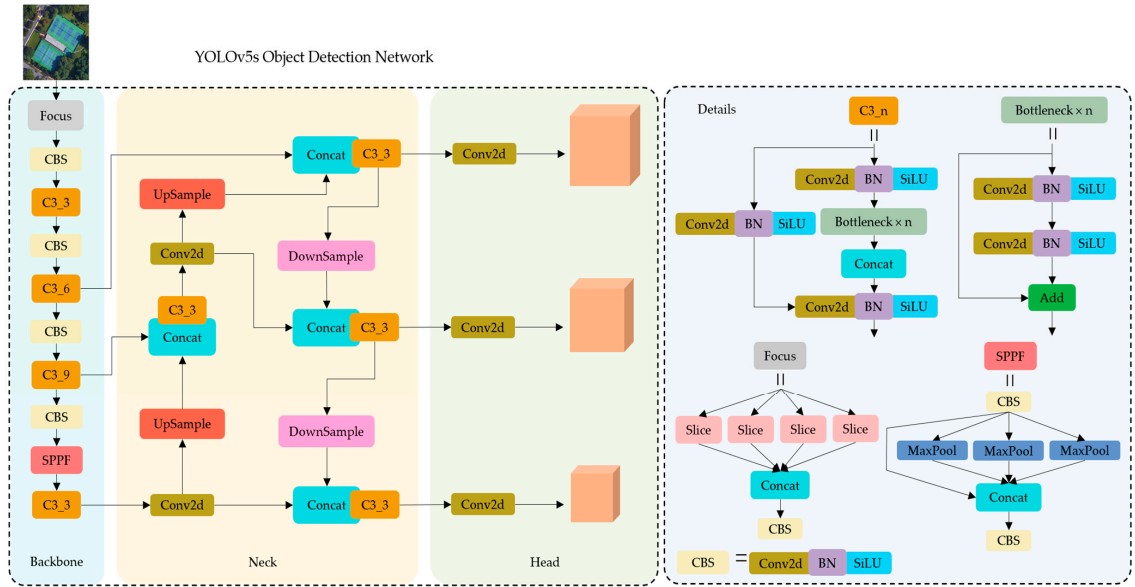

**Figure 2.** The network structure and details of YOLOv5s.

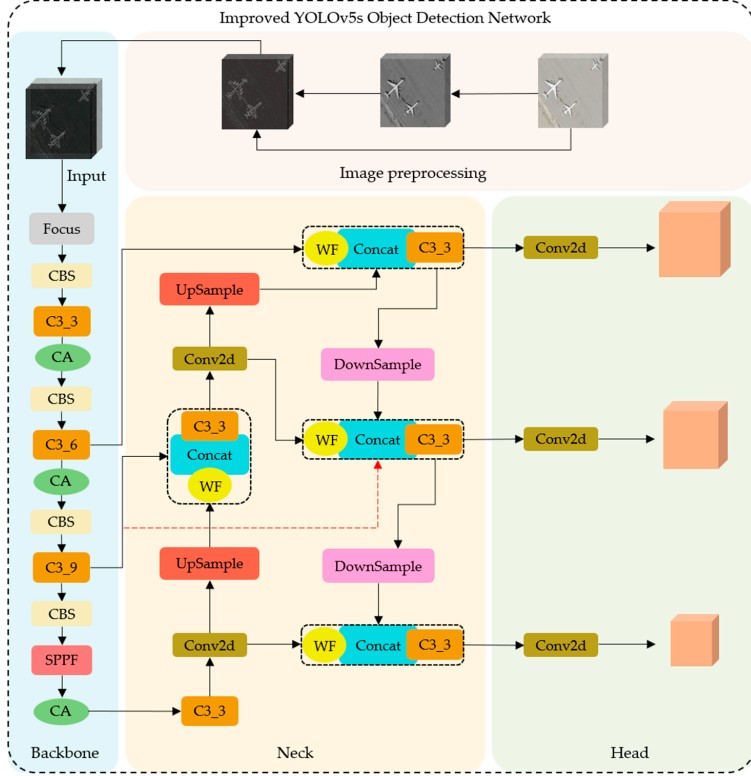

**Figure 3.** Diagram of the structure of the improved YOLOv5s object detection model.

### 3.3.1. Image Edge Preprocessing Based on Improved Gabor

Because UAV aerial photography is affected by the performance of the shooting equipment, flight height, light intensity, and so on, the aerial photography image often exhibits a complex background with blurry and dense target arrangement, which poses challenges for object detection. In order to reduce the difficulty of model detection and enhance the accuracy and speed of model detection, the image is preprocessed before inputting the color image into the model. In aerial photos of unmanned aerial vehicles, although the target color components are complex, the image target usually has corresponding edge representation. For this feature, we introduce an image processing method based on Gabor [58] for object edge enhancement. Because the gray value of the edge part of the object in the grayscale image changes significantly and contains high-frequency information, this method first converts the color picture into a grayscale image and then performs edge processing in the grayscale image.

In this paper, a filter bank composed of eight Gabor filters is used to enhance the edges of the target in the UAV aerial image according to its shape, which facilitates subsequent feature extraction. The Gabor function has been shown to perform well as a filter in texture recognition and detection of targets. In [59], the imaginary part of the Gabor function has good performance and robustness for edge detection of objects. The equations of the Gabor function and the imaginary part of the Gabor function are shown in Equations (1) and (2).

$$G(x,y) = \exp\left[-\frac{(x^2+y^2)}{2\sigma^2}\right] \cdot \exp[j\omega(x\cos\theta + y\sin\theta)] \tag{1}$$

$$G(x,y) = \exp\left[-\frac{(x^2+y^2)}{2\sigma^2}\right] \cdot \sin[\omega(x\cos\theta + y\sin\theta)] \tag{2}$$

The Gabor function is a Gaussian function modulated by a sinusoidal wave. $\sigma$ is the standard deviation of the Gaussian function along the coordinate axis. $\omega$ is the frequency value of the function in space, $\sigma \cdot \omega \leq 1$.

The use of traditional Gabor filters (TGFs) in [60] enhances the contour of the target in the image by segmenting the target area, which has a good effect. However, when the target has many directions, the processing method of TGFs is weak. In drone aerial photographs, the target always has any direction, which requires the combination of multiple Gabor filters from different directions in which the filter bank is closer to the edge of the target. Although a multi-directional and multi-scale filter bank allows the filtered image to have more complete information, it also increases computational costs, reduces the speed of image processing, and cannot meet the requirements of real-time performance. We have improved the traditional Gabor filter to address the above issues while meeting the needs of multi-directional and multi-scale filters. The improved Gabor filter no longer uses traditional fast Fourier transform to calculate at each pixel position but discretizes the infinite number of possible values into finite values during Gabor function operation and represents other values through quantization. According to [61], the quantization level is selected, and the value is selected based on the principle of proximity during the quantization process, with the intermediate value set to 0. In the Gabor function image, due to the positive and negative nature of the imaginary part of the function, the image is symmetrical along the coordinate axis, and the positive and negative quantization results are the same. We define the quantified value as $n_l$, where there is a total of $2n_l + 1$ values. $n_l$ is an important parameter for the discretization of convolution kernels, which can be understood as the dimension of discrete points. The purpose is to overcome the computational workload of traditional convolution kernels and improve the computational efficiency without obvious distortion in computational performance. To obtain the discrete Gabor values, we scatter the traditional continuous Gabor function values by defining the

positive and negative quantization values as $L_{positive}$ and $L_{negative}$ in quantization, and the positive and negative quantization equations are shown in (3) and (4), respectively.

$$L_{positive} = \frac{A}{2n_l + 1} \cdot 2k \tag{3}$$

$$L_{negative} = -\frac{A}{2n_l + 1} \cdot 2k \tag{4}$$

where $A$ is the amplitude of the Gabor function, and $k$ has a value range of $1 \sim n_l$. Then, the input image is convolved with a multi-scale and multi-directional filter, but the traditional Gabor filter convolution kernel divides the 2D plane into eight directions according to 360° for feature extraction. Here, due to the symmetric distribution of the selected convolution kernels, which have the same edge feature extraction ability in the opposite direction, we use the method of evenly dividing the 2D half-plane (i.e., 180°) to determine the direction of the convolution kernels as four. Due to the filter having the same function both in the four directions below the half plane and in the eight directions below the full plane, the time for feature extraction is halved, thereby improving detection efficiency. Considering the speed of the processing and meeting the multi-scale and multi-direction requirements, we use a two-scale and four-direction filter bank consisting of eight convolutional kernels. The two-scale here corresponds to the frequency or wavelength of the Gabor function. From the visual point of view, it determines the width of the Gabor convolution kernel texture. The convolution kernel should not be excessively large, as the edges are the local information of the image. An excessively large convolutional kernel can cause excessive coupling of pixel information and weaken the feature information of local pixels. After experimental verification, we chose a convolutional kernel of size $5 \times 5$. The $5 \times 5$ here represents the center selection $5 \times 5$ matrix of the discrete Gabor kernel. We chose the $5 \times 5$ matrix because the values outside the region have little impact on the convolution results, but the filtering characteristics of the values inside the matrix are still affected by it. The eight convolution kernels we used are shown in Figure 4. Among them, $\omega$ and $\theta$ are the frequency of Gabor convolution kernels in space and the direction of Gabor convolution kernel reinforcement, respectively.

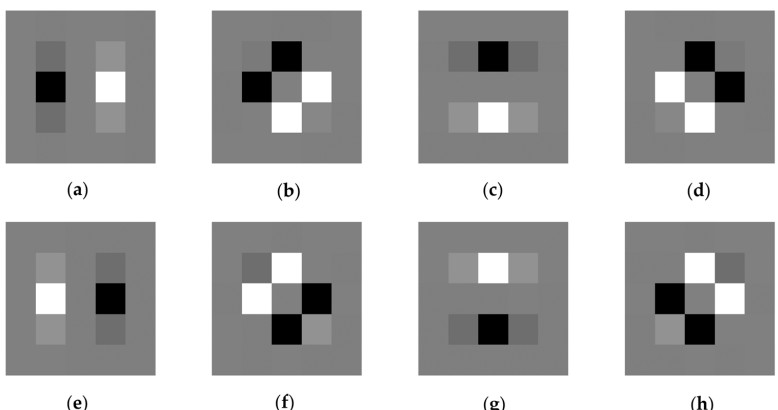

**Figure 4.** The parameters of the eight convolutional kernels: (**a**) $\omega = 0.3\pi$, $\theta = 0$; (**b**) $\omega = 0.3\pi$, $\theta = \pi j/4$; (**c**) $\omega = 0.3\pi$, $\theta = \pi j/2$; (**d**) $\omega = 0.3\pi$, $\theta = 3\pi j/4$; (**e**) $\omega = 0.5\pi$, $\theta = 0$; (**f**) $\omega = 0.5\pi$, $\theta = \pi j/4$; (**g**) $\omega = 0.5\pi$, $\theta = \pi j/2$; (**h**) $\omega = 0.5\pi$, $\theta = 3\pi j/4$.

Eight filters traverse convolution at each pixel location, as shown in Equation (5).

$$\phi_{\omega_i,\theta_j}(x,y) = I(x,y) \otimes \delta_{\omega_i,\theta_j}(x,y) \tag{5}$$

where $\phi_{\omega_i,\theta_j}(x,y)$ represents the filtered convolution value of the image, $I(x,y)$ represents the input image, $\delta_{\omega_i,\theta_j}(x,y)$ represents the filter bank, $\otimes$ represents the spatial domain

convolution calculation, and $\theta_j \in (0, \pi)$. The average segmentation of the 2D plane allows the filter to consider the edges in each direction during detection, which can robustly balance factors such as image flipping and rotation. After convolution, eight results will be obtained at each pixel position on the feature map. Among these eight results, the maximum value is selected through Equation (6), and the maximum value at each pixel is synthesized into a feature map with high-level semantics.

$$\phi'_{\omega,\theta}(x,y) = \max\left\{\phi_{\omega_i,\theta_j}(x,y), \ i = 0,1 \text{ and } j = 0,1,2,3\right\} \tag{6}$$

where $\phi'_{\omega,\theta}(x,y)$ is the composite result of the maximum value of each pixel, $i = 0, 1$, $j = 0, 1, 2, 3$. Each pixel value in $\phi'_{\omega,\theta}(x,y)$ is determined by the $\theta$ and $\omega$ parameters in the improved Gabor filter.

Figure 5 shows the results of the image filtered by the improved Gabor filter. (j) is a raw image selected for us with blur, noise, and similar backgrounds. In (a) to (h), eight filters were used to filter the image from eight directions and obtain feature maps. It can be seen that the edges of objects in the same direction as the filter have been significantly enhanced. The final composite image is shown in (i), and it is evident that the edges of the aircraft in this image have been truly enhanced. This indicates that the method we proposed is effective.

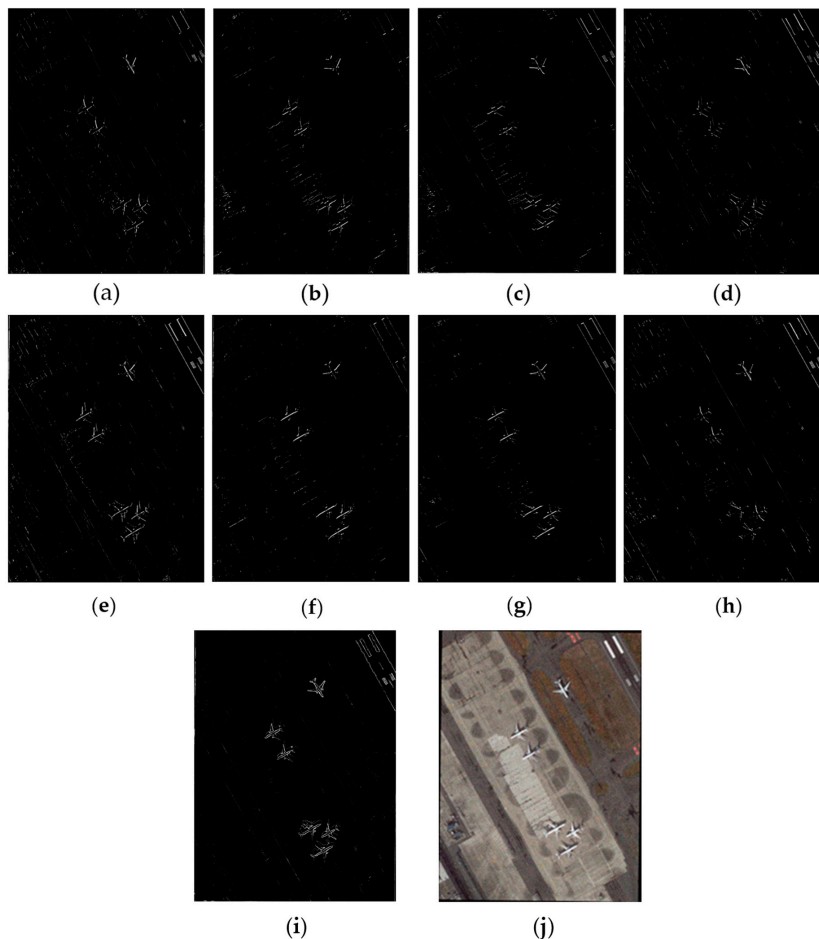

**Figure 5.** (**a–h**) The eight maximum output feature maps obtained after the sample is processed by the filter bank: (**a**) $\omega = 0.3\pi$, $\theta = 0$; (**b**) $\omega = 0.3\pi$, $\theta = \pi j/4$; (**c**) $\omega = 0.3\pi$, $\theta = \pi j/2$; (**d**) $\omega = 0.3\pi$, $\theta = 3\pi j/4$; (**e**) $\omega = 0.5\pi$, $\theta = 0$; (**f**) $\omega = 0.5\pi$, $\theta = \pi j/4$; (**g**) $\omega = 0.5\pi$, $\theta = \pi j/2$; (**h**) $\omega = 0.5\pi$, $\theta = 3\pi j/4$; (**i**) feature map synthesized from the maximum pixel values in eight directions; (**j**) original input diagram.

3.3.2. Coordinate Attention Mechanism

Due to the frequent occurrence of blurred edges, complex backgrounds, and small and concentrated targets in the drone aerial photos input into the model, the model backbone may obtain unsatisfactory results in feature extraction, which poses difficulties for subsequent feature fusion and accurate prediction information output. In view of this, researchers often introduce a method to improve feature extraction performance: the attention mechanism [45–49]. The attention mechanism can flexibly capture the connections between global and local information, making the network more interested in the target area and less focused on other areas. By putting more weight on the object region, we can obtain more detailed information on the object and suppress other useless information at the same time, so as to improve the efficiency of feature extraction of the model.

Currently, there are various types of attention mechanisms with varying performance. For example, the Non-Local Network [62] ignores the distance between any two positions when performing non-local operations and directly captures long-range dependencies by calculating the interaction between the two positions. However, the network has only independent positional attention modules and no channel attention mechanism. When the feature map size is large, the network consumes a large amount of memory and computing resources to compare each point in the feature map. SENet [63] has a stronger feedback ability by focusing on which layers of information are at the channel level through squeezing and motivating, but attention is not expressed in the spatial dimension; that is, position information is not expressed in the visual target. CBAM [64] introduces attention mechanisms in both channel and spatial dimensions with the aim of embedding position information into channel attention through global pooling. However, this module does not capture spatial information of different scales to enrich features, and spatial attention only considers local region information, which cannot allow the establishment of long-distance dependencies.

In contrast, we introduce a flexible and efficient plug-and-play coordinate attention (CA) mechanism that embeds location information into channel attention, enabling the backbone CSPDark network to avoid excessive computational complexity by obtaining information from a larger area. Traditional attention mechanisms can cause a loss of positional information when performing 2D global pooling, while CA transforms global pooling into two encoding operations of 1D vectors. Specifically, for input X ($C \times H \times W$), pooling kernels ($H$, 1) and (1, $W$) are used to encode horizontal and vertical features, respectively, to obtain the output of the $c$-th dimensional features, as shown in Equations (7) and (8).

$$z_c^h(h) = \frac{1}{W} \sum_{0 \leq i < W} x_c(h, i) \tag{7}$$

$$z_c^w(w) = \frac{1}{H} \sum_{0 \leq i < H} x_c(j, w) \tag{8}$$

where $z_c^h(h)$ is the output of the $c$-th dimensional feature at height $h$, and $z_c^w(w)$ is the output of the $c$-th dimensional feature at width $w$. The above formula integrates features from different directions and outputs a pair of directional feature maps. Compared to the compression method of global pooling, this allows the attention module to capture long-distance relationships in one direction while retaining spatial information in the other direction, helping the network to more accurately locate targets. Therefore, location information can be saved in the generated attention map. Then, the outputs of Formulas (7) and (8) are concatenated using $1 \times 1$ convolutional kernel, BN, and nonlinear activation for feature transformation, as shown in Equation (9).

$$f = \delta\left(F_1\left(\left[z^h, z^w\right]\right)\right) \tag{9}$$

where $f \in \mathbb{R}^{C/r \times (H+W)}$ is the intermediate feature that contains both horizontal and vertical spatial information, $r$ is the reduction factor, and $\delta$ is the nonlinear activation function. There is no intense fusion of these two features. The main purpose of concatenating is to unify BN operations. $f$ is then divided into two separate features, $f^h \in \mathbb{R}^{C/r \times H}$ and $f^w \in \mathbb{R}^{C/r \times W}$, using the other two $1 \times 1$ convolutional kernel and sigmoid function transform features so that their dimensions are consistent with input X ($C \times H \times W$), such as Formulas (10) and (11).

$$g^h = \sigma\left(F_h\left(f^h\right)\right) \tag{10}$$

$$g^w = \sigma(F_w(f^w)) \tag{11}$$

The multiplication of output $g^h$ and $g^w$ into a weight matrix, as shown in Equation (12), is used to calculate coordinate attention block output to emphasize the representation of the area of interest.

$$g_c(i,j) = x_c(i,j) \times g_c^h(i) \times g_c^w(j) \tag{12}$$

Each weight of the coordinate attention block contains channel information, horizontal spatial information, and vertical spatial information, which can help the network to more accurately locate and identify the target area information. This method is flexible and lightweight and can easily be incorporated into existing classical mobile networks, such as CSPDarkNet, ShuffleNet, etc., to improve the performance of feature representation. The principle of the coordination attention mechanism is shown in Figure 6.

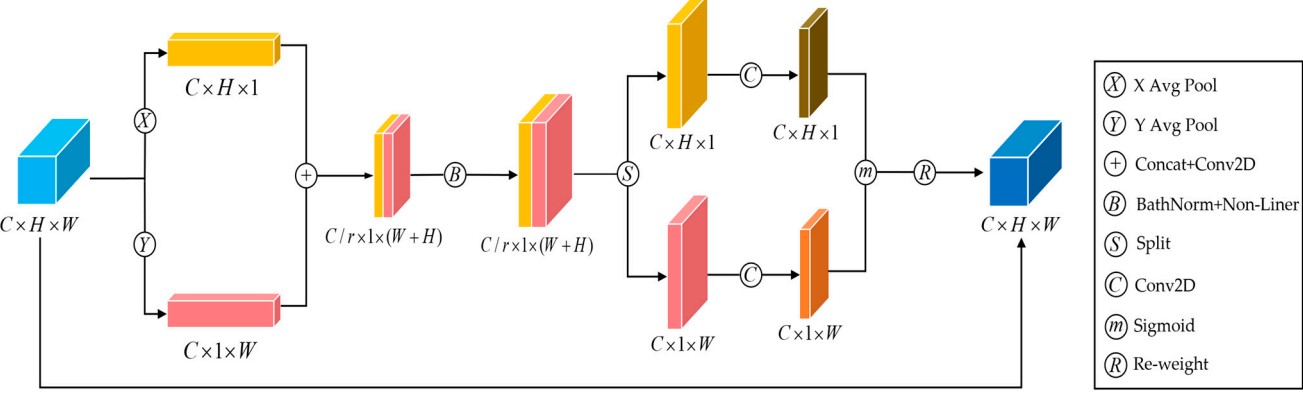

**Figure 6.** The principle diagram of the coordinate attention mechanism.

### 3.3.3. Bidirectional Feature Fusion Network

The neck network part of YOLOv5 consists of a feature pyramid network structure, PANet, aimed at fusing different scale features input from the backbone. Although the original FPN structure improved the high-level semantic information in the prediction feature map, much of the location information in the underlying feature map is lost. Therefore, PANet adds a bottom-up channel on the basis of the FPN to transmit the location information to the prediction feature map such that the prediction feature map simultaneously has both high semantic and location information. This can greatly improve the accuracy of object detection. The specific structure of PANet is shown in Figure 7.

Although PANet treats input features of different scales equally during feature fusion, wherein features of different resolutions are directly added for output during fusion, in reality, however, their contributions to the final output are different. In order to balance the contributions of each feature layer, this paper proposes a new feature fusion method: Bidirectional Feature Pyramid Network (BiFPN). This involves modifying the PANet structure in YOLOv5 to a BiFPN structure, proportionally weighting features of different sizes to balance the information of each input feature. This network can achieve bidirectional fusion of deep and shallow features from top to bottom and from bottom to top, enhanc-

ing the transmission of feature information between different network layers, which will significantly improve the detection accuracy and performance of the YOLOv5 algorithm.

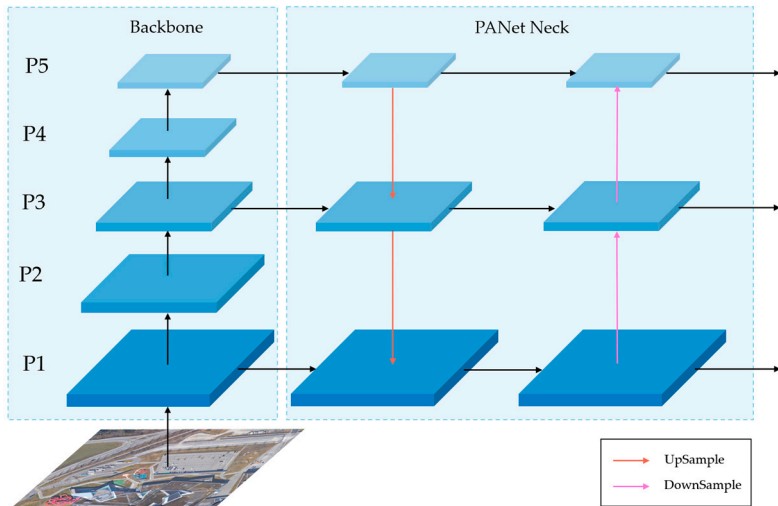

**Figure 7.** The network structure diagram of PANet.

Specifically, the following improvements in design have been made in the BiFPN structure. Firstly, nodes with only one input edge and no feature fusion are removed due to their lower contribution to feature fusion in the feature network. This allows the network to be simplified without affecting its performance. Secondly, we add an additional feature branch between the original input and output nodes in the same layer. This will not increase the computational cost too much and can also allow the integration of more features. Finally, due to the different resolutions of multi-scale feature maps, their contributions to fusion as inputs vary. Therefore, we balance different feature layers through weighting to achieve a deep fusion of feature layers. However, traditional feature fusion structures do not distinguish between input feature maps during concatenate or shortcut connections but simply stack or add them. We use the middle layer as an example to demonstrate the relationship between each feature layer, as shown in Equations (13) and (14).

$$P_3^{Td} = Conv(P_3^{In} + Resize(P_5^{Out})) \tag{13}$$

$$P_3^{Out} = Conv(P_3^{Td} + Resize(P_1^{Out})) \tag{14}$$

where $P_3^{Td}$ is the intermediate feature of the third layer, and $P_3^{Out}$ is the output feature of the third layer. For the shortcomings exposed by this traditional feature fusion method, we propose a feature fusion mechanism that weights input features, adjusting the contribution of different feature maps through trainable weights. Considering the differences in training performance, we chose the fast normalization fusion method, as shown in Equation (15). This method reduces the weight range to 0~1, which results in a fast training speed and high efficiency.

$$O = \sum_i \frac{w_i}{\varepsilon + \sum_j w_j} I_i \tag{15}$$

where $w$ is the weight learned by the feature, and $I_i$ is the input feature map, taking the stable numerical coefficient $\varepsilon = 0.0001$. The fusion process of the two features in the middle layer is shown in Equations (16) and (17).

$$P_3^{Td} = Conv\left(\frac{w_1 p_3^{In} + w_2 Resize(P_5^{In})}{w_1 + w_2 + \varepsilon}\right) \tag{16}$$

$$P_3^{Out} = Conv\left(\frac{w_1' p_3^{In} + w_2' P_3^{Td} + w_3' Resize(P_1^{Out})}{w_1' + w_2' + w_3' + \varepsilon}\right) \tag{17}$$

where *Conv* is a deep separable convolution operation, and *Resize* is an upsampling or downsampling operation. The BiFPN network structure is shown in Figure 8.

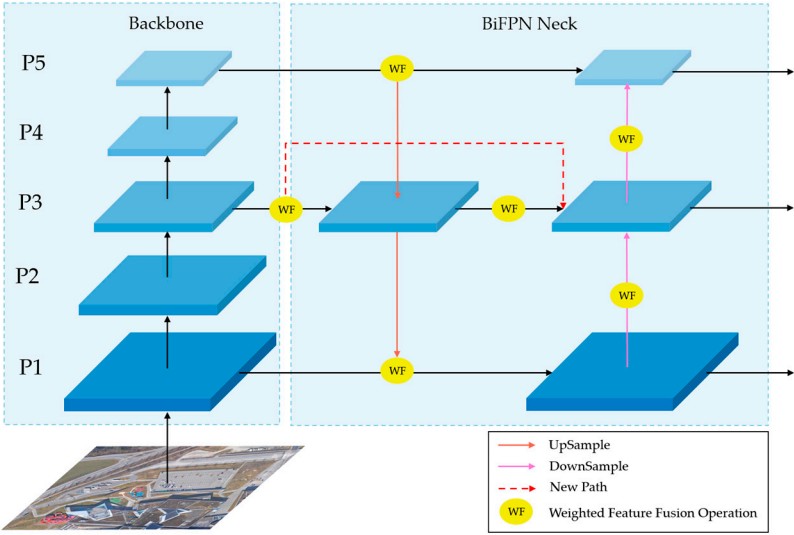

**Figure 8.** The network structure diagram of BiFPN.

## 4. Experiments and Results

In this section, firstly, we collected three UAV aerial image datasets, UCAS-AOD [65], DOTA-V1.0 [66], and VisDrone2019 [67], and analyzed the data from each dataset. Secondly, in order to maximize the learning ability of the model, we integrated the above three datasets into a new dataset called LSDUVD through data augmentation. We analyzed the data from the above four datasets and explained the advantages of LSDUVD based on their proportion. Once again, to verify the effectiveness of our improved method, we conducted a series of experiments on UCAS-AOD, DOTA-V1.0, VisDrone2019, and LSDUVD. The experiment is divided into three parts. (1) On the dataset LSDUVD, training experiments were conducted on the original model, corresponding models of each improved module, and the overall improved model we proposed (referred to as YOLOv5s, YOLOv5s-Gabor, YOLOv5s-CA, YOLOv5s-BiFPN, YOLOv5s-Improved). Based on the training loss–epochs curve obtained after the experiment, the convergence situation was analyzed to demonstrate the advantages of our proposed model. (2) We conducted ablation experiments and analyzed the results of $AP_s$, $AP_m$, $AP_l$, $mAP@0.5$, and $mAP@[0.5:0.95]$ of each model in (1) to verify that our proposed method can effectively classify and locate multiple types of targets in drone aerial images. (3) To further highlight the effectiveness of our proposed method, we calculated precision–recall rates for the above five models on UCAS-AOD, DOTA-V1.0, VisDrone2019, and LSDUVD. By observing the precision–recall curves, we found that our proposed method is effective in improving detection accuracy. (4) Comparing our method with several popular object detection networks, the detection results show that our method has good detection accuracy in complex drone aerial images.

### 4.1. Experimental Environment and Training Parameter Settings

The parameters of the hardware and software environment we used during the experiment are shown in Table 1. For example, the CPU is Intel Core i7-9700k, the GPU is NVIDIA GeForce RTX 3090Ti, the operating system is Ubuntu18.04 LTS 64-bits, the deep learning framework is PyTorch1.9.1, CUDA 10.2 is the GPU accelerator, etc.

**Table 1.** Software and hardware parameter configuration in the experimental environment.

| Parameter | Configuration |
|---|---|
| Integrated development environment | PyCharm |
| Scripting language | Python3.8 |
| Deep learning frame | PyTorch1.9.1 |
| CPU model | Intel Core i7-9700k |
| Operating system | Ubuntu18.04 LTS 64-bits |
| GPU model | NVIDIA GeForce RTX 3090Ti |
| GPU accelerator | CUDA 10.2 |
| Neural network accelerator | cuDNN7.6.5 |

The training experiment parameters are shown in Table 2. A stochastic gradient descent (SGD) optimizer with a momentum of 0.937 was used to train the network. The initial learning rate was 0.001, the weight decay was 0.0005, and the batch size was set to 32. A total of 1600 epochs were trained.

**Table 2.** Configuration of training experimental parameters.

| Parameter | Configuration |
|---|---|
| Neural network optimizer | SGD |
| Learning rate | 0.001 |
| Training epochs | 1600 |
| Momentum | 0.937 |
| Batch size | 32 |
| Weight decay | 0.0005 |

*4.2. Dataset Preparation*

The overall improved YOLOv5 model was trained on our integrated dataset. We named the integrated dataset LSDUVD (Large-Scale Dataset Based on UCAS-AOD, Vis-Drone2019, and DOTA-V1.0), which consists of data-enhanced UCAS-AOD, VisDrone2019, and DOTA-V1.0. The DOTA-V1.0 dataset released by Wuhan University is a large-scale aerial image dataset with targets of different proportions, directions, and shapes. The dataset consists of 15 common categories such as vehicles, stadiums, boats, and over-passes, including 2806 images and 188,282 samples with sizes ranging from $800 \times 800$ to $4000 \times 4000$ pixels. The annotation method for each sample instance is the bounding box of any quadrilateral determined by four points. VisDrone2019 was produced by Tianjin University, in which various drone lenses are used to capture images in different scenes and environments. This dataset contains 10 types of objects, including pedestrians, cars, and tricycles, with a total of 10,209 images. UCAS-AOD is a high-definition aerial photography dataset released by the University of the Chinese Academy of Sciences. Most of the objects in this dataset are small targets of two categories: cars and planes. There are 510 images of cars, totaling 7114 samples, and 1000 images of planes, totaling 7482 samples. In this work, the samples were enriched through data augmentation methods such as vertical flipping, horizontal flipping, random cropping, and cutout, improving the model's generalization ability and robustness. Figure 9 shows some original and data-enhanced images.

We used the LabelImg tool to manually annotate images using traditional horizontal bounding boxes. For situations of severe occlusion, targets with an occlusion area greater than 90% and an edge area less than 15% are not marked. We define targets smaller than $32 \times 32$ as small targets, targets between $32 \times 32$ and $96 \times 96$ as medium targets, and targets larger than $96 \times 96$ as large targets. Based on the training requirements, we selected 4000, 15,000, and 3000 images from three enhanced datasets to form the LSDUVD dataset. As shown in Figure 10, we used visual methods to calculate the proportion of sample types and instance numbers for the four datasets mentioned above. The graph intuitively reflects that LSDUVD is a large-scale dataset rich in types and quantities, which ensures that the model has efficient detection ability after training.

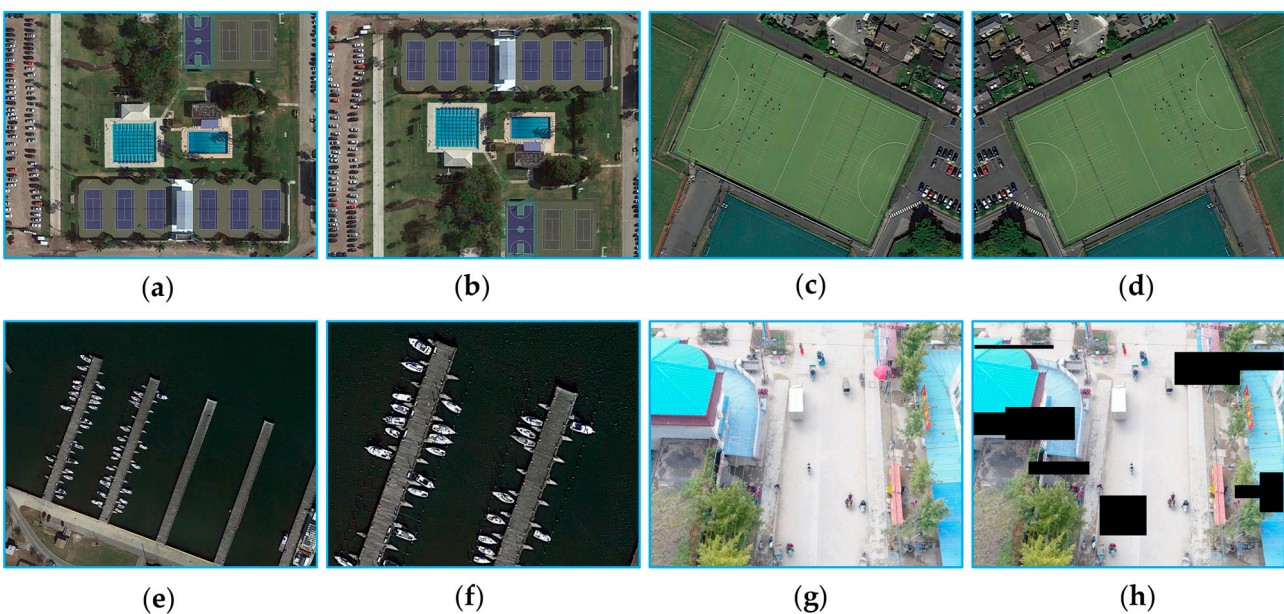

**Figure 9.** (**a,c,e,g**) Original images. (**b,d,f,h**) Images after vertical flipping, horizontal flipping, random cropping, and cutout, respectively.

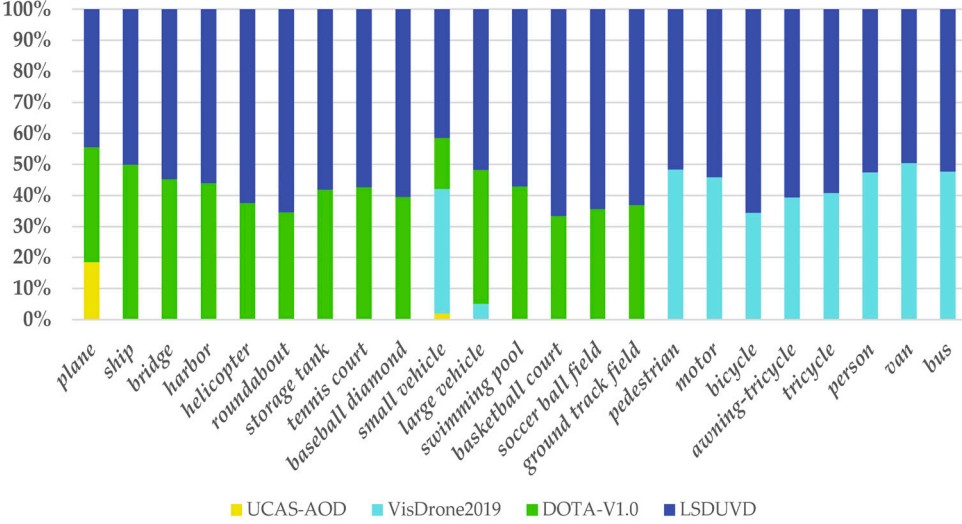

**Figure 10.** The above figure shows the statistical results of the proportion of sample types and instance numbers for UCAS-AOD, VisDrone2019, DOTA-V1.0, and LSDUVD datasets.

### 4.3. Performance Evaluation Indicators

Due to the large receptive field of UAV aerial images, there are many types of objects, complex backgrounds, and large differences in target size in the images, which inevitably increases the difficulty of detection and may increase the rates of missed detection and false detection of models. In order to fully demonstrate the improved detection performance of the model, this article uses four evaluation indicators to measure the performance of each model in the experiment: recall rate ($R$), precision ($P$), average precision ($AP$), and mean average precision ($mAP$). The formulas for defining recall rate and accuracy are shown in (18) and (19).

$$R = \frac{TP}{TP + FN} \times 100\% = \frac{TP}{all \ ground \ truth} \times 100\% \tag{18}$$

$$P = \frac{TP}{TP + FP} \times 100\% = \frac{TP}{all \ \text{detect}} \times 100\% \tag{19}$$

where $TP$ is true positive, which refers to the positive sample data correctly detected by the model; $FN$ is false negative, which refers to the correct sample data that the model did not detect; $FP$ is false positive, which refers to the positive sample data detected by model errors. The general way to distinguish $TP$ and $FP$ is based on setting the threshold of $IOU$, where $IOU$ is the ratio of the intersection area between the predicted box area and the real box area, as shown in Equation (20).

$$IOU = \frac{area(B_{\det} \cap B_{get})}{area(B_{\det} \cup B_{get})} \times 100\% \tag{20}$$

where $B_{\det}$ refers to the predicted box area, $B_{get}$ refers to the actual box area, $area(B_{\det} \cap B_{get})$ refers to the intersection area of two boxes, and $area(B_{\det} \cup B_{get})$ refers to the union area of two boxes. In this article, $area(B_{\det} \cap B_{get})$ refers to the intersection area of two boxes. We use the ratio of $IOU$ greater than 0.5 to denote $TP$ and less than 0.5 to denote $FP$.

Although the greater the recall and precision of the model the better, the change in recall and precision is a mutually constraining and inversely proportional relationship. In order to find the best balance between them, we introduced the $AP$ value, which is the average precision. We use the $P - R$ curve to calculate the $AP$ value. The precision in the $P - R$ curve is the vertical axis, the recall rate is the horizontal axis, and the area formed between the curve and the coordinate axis is the $AP$ value. Because having multiple average precision values causes difficulties in intuitively measuring the entire model, we chose $mAP$ to represent the comprehensive detection performance of a model. $mAP$ refers to the average value of $AP$ for all classes. The calculation formulas for $AP$ and $mAP$ are shown in (21) and (22).

$$AP = \int_0^1 P(R)dR \times 100\% = \sum_{k=0}^{n} P(k)R(k) \times 100\% \tag{21}$$

$$mAP = \frac{1}{Q} \sum_{q=1}^{Q} AP(q) \times 100\% \tag{22}$$

where $P(R)$ represents the precision value on the $P - R$ curve with a recall rate of $R$; $k$ represents a certain cut-off position of precision; $P(k)$ and $R(k)$ represent the values of precision and recall rate at position $k$, respectively; $n$ represents the range of $k$ points; $q$ represents the category of detected objects; and $Q$ represents the total number of detected object categories.

### 4.4. Experimental and Results Analysis

4.4.1. Model Training Experiment

In this section, the performance of our proposed model is verified through a training loss experiment. Firstly, the experimental parameters were set. Secondly, during training, we trained the five models YOLOv5s, YOLOv5s-Gabor, YOLOv5s-CA, YOLOv5s-BiFPN, and YOLOv5s-Improved on the training set of LSDUVD. The loss–epoch change curve after training is shown in Figure 11. The figure shows that the loss trends of the five models tend to stabilize with the increase in epoch values. Specifically, YOLOv5s-Improved has a faster convergence speed and the best convergence effect compared to other models. It tends to saturate after 800 epoch values and eventually converges to around 0.5. However, the loss function curves of YOLOv5s, YOLOv5s-Gabor, YOLOv5s-CA, and YOLOv5s-BiFPN all stabilized between 1000 and 1200 epochs, and their loss function values converged to around 3.0, 2.2, 1.5, and 1.3, respectively. From the above data, it can be seen that the loss for YOLOv5s-Improved is approximately 2.5 lower than for the original YOLOv5s, indicating a significant improvement in the performance of our proposed model. From the graph, it can be seen that the loss epoch curves of the five models have similar trends. This is because our method is optimized based on YOLO and without fundamentally redesigning

the network structure. Therefore, further in-depth research on network structure is of great significance.

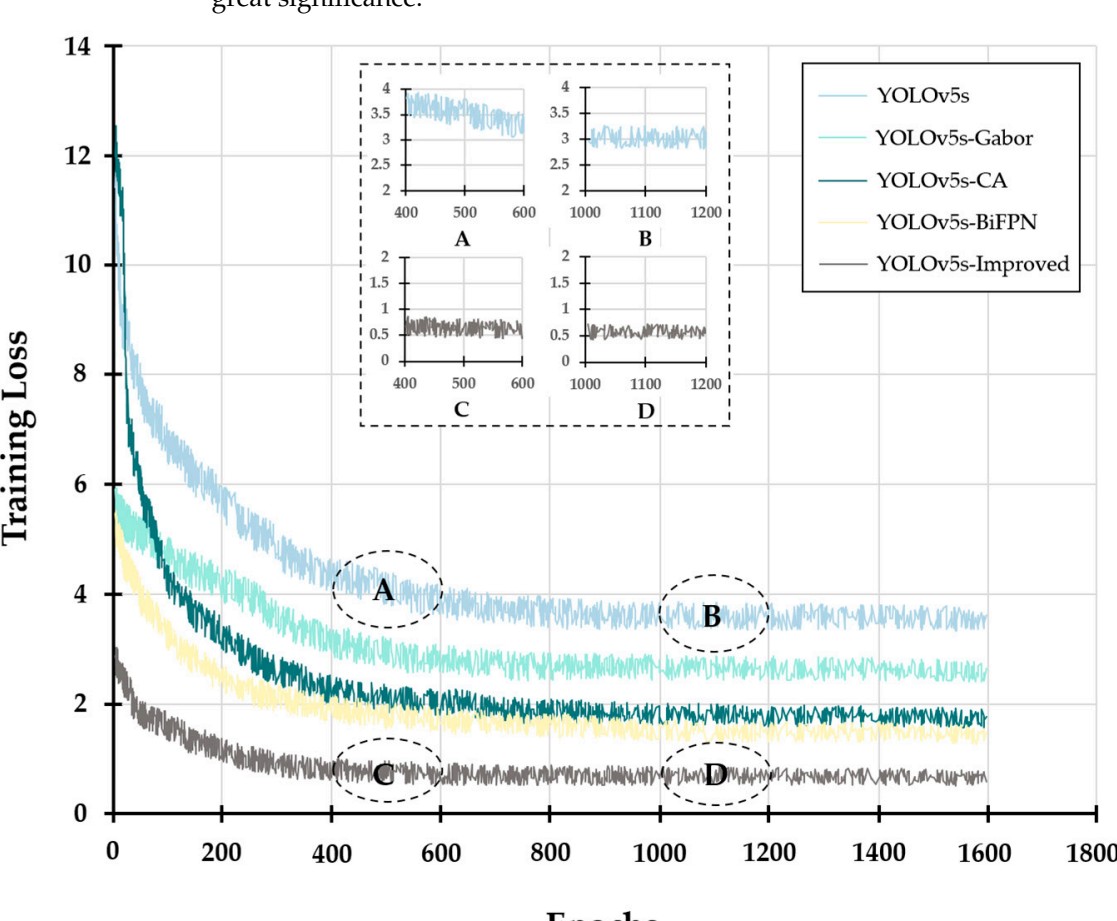

**Figure 11.** Training loss–epochs curve.

In order to further analyze the data, we selected YOLOv5s and YOLOv5s-Improved for enlarged display on data for 400–600 and 1000–1200 epochs, as shown in Figure 11 A–D. From the graph, it can be seen that within the same epoch interval range, such as the comparison between A and C, as well as the comparison between images B and D, our method results in a curve with a smaller vibration amplitude. This indicates that, with the same data support, the improved YOLOv5s network has higher robustness. Through a comparative analysis of A and B, as well as C and D, the improved method significantly reduces data fluctuations, indicating that the robustness of the improved method is strengthened with the accumulation of knowledge. This also strengthens the certainty of the network as it moves toward application.

### 4.4.2. Ablation Experiment

To highlight the significant detection performance of our algorithm model, we conducted an ablation experiment on each improved module. The experimental results are shown in Table 3. The table shows the detection results for the YOLOv5s baseline model and the models with the Gabor module, CA module, BiFPN module added, as well as the overall improved model. $AP_s$ represents the average detection precision of small targets, $AP_m$ represents the average detection precision of medium targets, $AP_l$ represents the average detection precision of large targets, $mAP@0.5$ represents the average value of all categories $AP$ when the $IOU$ threshold is set to 0.5, and $mAP@[0.5 : 0.95]$ represents the average $mAP$ value of the $IOU$ threshold starting from 0.5 in steps of 0.05 to 0.95.

**Table 3.** The ablation results of each improved module, with data expressed in percentage (%). The best results are highlighted in bold.

| Baseline | Gabor | CA | BiFPN | Dataset | $AP_s$ | $AP_m$ | $AP_l$ | $mAP@0.5$ | $mAP@[0.5:0.95]$ |
|---|---|---|---|---|---|---|---|---|---|
| YOLOv5s | | | | LSDUVD | 80.4 | 82.4 | 85.8 | 81.9 | 64.3 |
| | √ | | | LSDUVD | 82.1 | 83.2 | 86.7 | 83.5 | 66.1 |
| | | √ | | LSDUVD | 83.7 | 83.7 | 86.1 | 84.2 | 67.2 |
| | | | √ | LSDUVD | 81.1 | 84.0 | 87.5 | 84.5 | 67.7 |
| | √ | √ | √ | LSDUVD | **85.3** | **88.1** | **90.2** | **85.7** | **69.5** |

By analyzing the data in the table, it can be seen that compared with the detection performance of the original YOLOv5, firstly, the Gabor module we introduced improves $AP_s$, $AP_m$, and $AP_l$ by 1.7%, 0.8%, and 0.9%, respectively, while $mAP@0.5$ and $mAP@[0.5:0.95]$ improved by 1.6% and 1.8%, respectively. It can be seen that the edge reinforcement characteristics of this module have a positive impact on the model detection performance. Secondly, the introduction of the CA mechanism in the main part of the model makes the model more able to overcome the problem of insufficient feature information extraction of small targets. It has increased $AP_s$ by 3.30%, significantly improving the detection precision of small targets. At the same time, the $AP$ and $mAP$ values of medium to large targets have also improved. Thirdly, the use of bidirectional feature fusion structure BiFPN has improved the detection performance of various types of targets, with $AP_s$, $AP_m$, and $AP_l$ increasing by 0.7%, 1.6%, and 1.7%, and $mAP@0.5$ and $mAP@[0.5:0.95]$ increasing by 2.6% and 3.4%, respectively. It can be seen that the BiFPN module is suitable for detecting medium to large targets. Finally, the combination of the Gabor module, CA module, and BiFPN module resulted in an increase in $AP_s$, $AP_m$, and $AP_l$ by 4.9%, 5.7%, and 4.4%, respectively, while $mAP@0.5$ and $mAP@[0.5:0.95]$ increased by 3.8% and 5.3%, respectively. Based on the above detection results, it can be seen that combining the three modules on YOLOv5s results in the best $AP$ and $mAP$ values, and the detection performance of targets at all scales is greatly improved. This indicates that each module can operate in coordination and work together on YOLOv5s, which also indicates that the method proposed in this article is reasonable and efficient.

### 4.4.3. Precision–Recall Rate Experiment

To verify the performance of our method, we generated precision–recall rate statistics, and the statistical results are shown in Figure 12. We mainly verify two aspects: firstly, the impact of the size of the database and the number of categories on precision–recall rate, and secondly, the impact of our improved module on the results. In these four statistical results, each result corresponds to four independent databases, including DOTA-V1.0, VisDrone2019, UCAS-AOD, and LSDUVD. In order to verify the impact of introducing modules on the algorithm, we conducted comparative experiments on five methods in each graph. Among them, ⑤ represents YOLOv5s, ④ represents the statistical results of introducing the Gabor preprocessing module, and we named this model YOLOv5s-Gabor, ③ and ② represent the experimental results after separately introducing CA and BiFPN modules, while curve ① represents the overall performance of our improved algorithm, and we call them YOLOv5s-CA, YOLOv5s-BiFPN, and YOLOv5s-Improved, respectively.

From the statistical results, it can be seen that among these four databases, UCAS-AOD has the worst performance in terms of the precision–recall rate. After our analysis, we found that the UCAS-AOD database has a smaller sample size and fewer categories for classification. In VisDrone2019 and DOTA-V1.0, the precision–recall rate showed better statistical results. From the experimental results, our method performs the best on dataset LSDUVD. This indicates that the good performance of the network requires a large amount of data resources as information support.

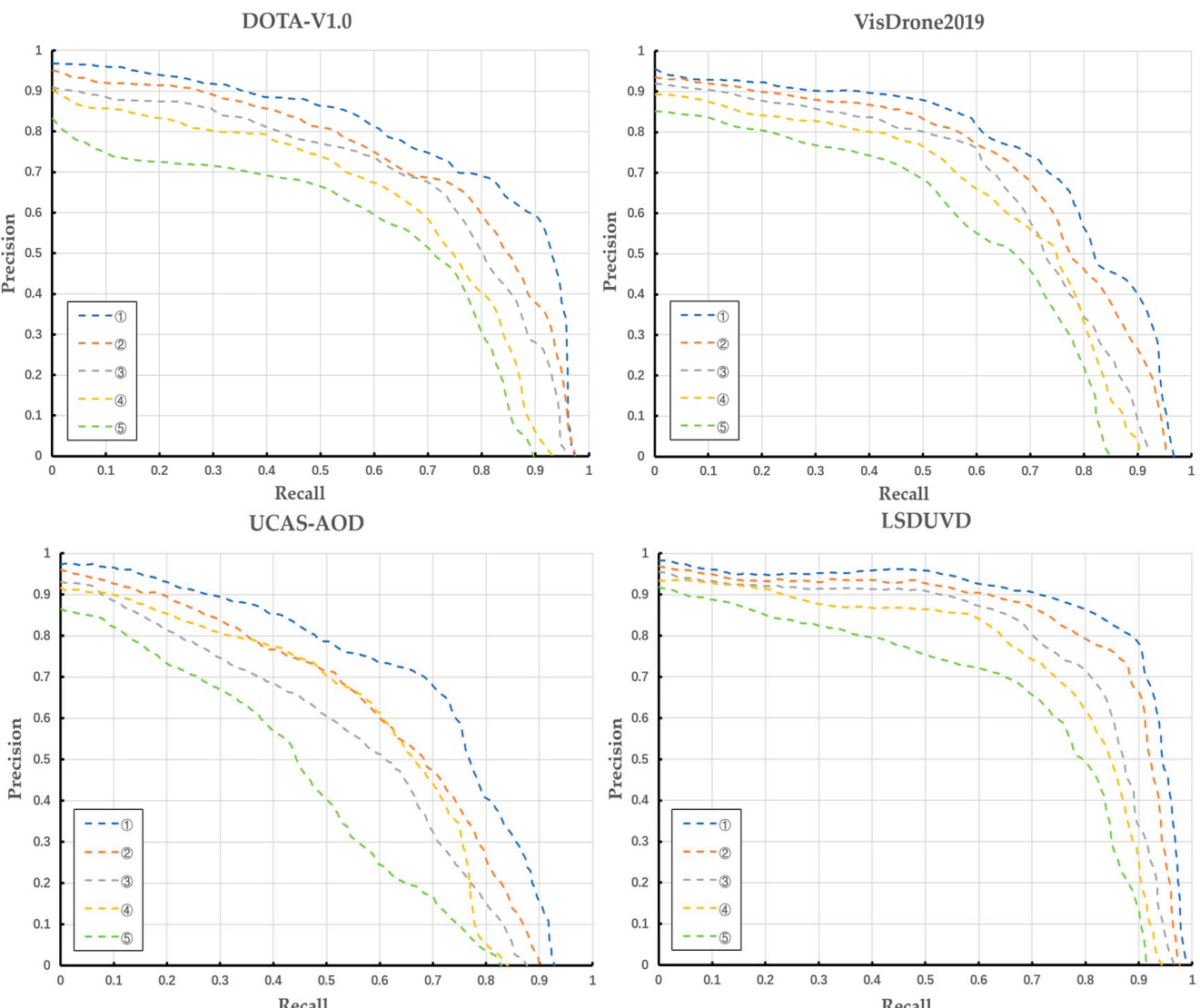

**Figure 12.** The precision–recall curves of the five methods.

From the perspective of method, the baseline model YOLOv5s has the worst precision–recall performance compared with the four models YOLOv5s-Gabor, YOLOv5s-CA, YOLOv5s-BiFPN, and YOLOv5s-Improved. The detection performance of the model combined with each improved module has been improved to varying degrees, and the detection effect of YOLOv5s-CA and YOLOv5s-BiFPN is better than that of YOLOv5s-Gabor in general. YOLOv5s-Improved has the best performance of the precision–recall rate among all models in this experiment, which further shows that the overall improved YOLOv5s model integrating Gabor, CA, and BiFPN is efficient in the object detection task of UAV aerial images.

### 4.4.4. Comparison of Several Object Detection Methods

To further validate the efficient detection performance of our proposed method, we compared several common object detection algorithms, such as SSD [22], FMSSD [68], Faster-RCNN [26], RetinaNet [69], YOLOv3 [30], YOLOv4 [31], and YOLOv5s. The detection experiment was conducted on the dataset LSDUVD, and the *AP* (%) and *mAP* (%) results of each target are shown in Table 4. We have abbreviated the target names in the dataset as follows: PE, plane; SP, ship; BE, bridge; HR, harbor; HP, helicopter; RT, roundabout; ST, storage tank; TC, tennis court; BD, baseball diamond; SV, small vehicle; LV, large vehicle; SL, swimming pool; BC, basketball court; SBF, soccer ball field; GTF, ground

track field; PN, pedestrian; MT, motor; BI, bicycle; ATE, awning tricycle; TE, tricycle; PR, person; VN, van; BS, bus. By analyzing the *AP* and *mAP* values in the table, overall, our method has the highest *mAP* (81.8%) in all categories of object detection, which is 5.4% higher than other models with the highest *mAP*. Specifically, in the detection of large and medium-sized targets such as soccer ball fields, basketball courts, baseball diamonds, roundabouts, large vehicles, swimming pools, etc., our model is not significantly different from other models in having the best detection results, as they all have high detection *AP* values. This indicates that our method has good detection accuracy in the detection of medium-sized and large targets. In terms of small object detection, compared with the best detection results of the other seven methods in Table 4, our model increased the *AP* values of small objects such as planes, ships, storage tanks, small vehicles, and pedestrians by 1.6%, 5.3%, 7.3%, 7.4%, and 0.6%, respectively. This indicates that our model has the best detection ability for small objects, especially for dense small objects such as ships, storage tanks, and small vehicles, which have more obvious detection effects.

**Table 4.** The *AP* (%) and *mAP* (%) results of our method and several common methods on the LSDUVD dataset. The best results are presented in bold.

| Target Category | Methods (*AP* (%)) | | | | | | | |
|---|---|---|---|---|---|---|---|---|
| | SSD | FMSSD | Faster-RCNN | RetinaNet | YOLOv3 | YOLOv4 | YOLOv5s | Ours |
| PE | 81.9 | 89.1 | 75.7 | 87.2 | 90.5 | 93.1 | 91.5 | **94.7** |
| SP | 69.2 | 76.9 | 75.2 | 71.7 | 82.7 | 82.3 | 80.8 | **87.6** |
| BE | 59.2 | 68.2 | 55.0 | 67.0 | 77.2 | 68.1 | 73.2 | **79.6** |
| HR | 64.5 | 72.4 | 67.3 | 71.8 | 77.8 | 81.9 | 83.0 | **85.7** |
| HP | 71.8 | 70.2 | 68.8 | 67.7 | 82.0 | **83.8** | 66.9 | 64.9 |
| RT | 73.5 | 67.5 | 79.2 | 56.2 | 71.6 | 67.7 | 79.2 | **82.9** |
| ST | 74.5 | 73.7 | 68.4 | 70.2 | 60.7 | 68.5 | 76.9 | **84.2** |
| TC | 89.6 | 90.7 | 85.6 | **95.3** | 94.3 | 94.6 | 93.0 | 94.5 |
| BD | 81.3 | 81.5 | 77.5 | **90.6** | 64.0 | 67.1 | 73.4 | 89.5 |
| SV | 63.0 | 79.2 | 60.8 | 70.7 | 70.3 | 70.8 | 75.2 | **86.6** |
| LV | 59.4 | 73.6 | 79.0 | 83.0 | 79.9 | 81.1 | 88.9 | **90.2** |
| SL | 65.6 | 80.6 | 66.7 | 70.7 | **88.6** | 74.0 | 71.0 | 76.6 |
| BC | 72.2 | 82.7 | 74.2 | **88.5** | 72.8 | 65.9 | 83.6 | 87.4 |
| SBF | 74.5 | 78.7 | 58.4 | **87.9** | 70.7 | 72.9 | 77.3 | 86.2 |
| GTF | **79.9** | 67.9 | 74.7 | 79.7 | 59.7 | 62.8 | 73.6 | 77.3 |
| PN | 58.5 | 70.3 | 64.5 | 72.1 | 60.7 | 61.7 | 64.0 | **72.7** |
| MT | 65.5 | 78.1 | 63.1 | 67.5 | 77.1 | **80.2** | 76.3 | 78.5 |
| BI | 70.2 | 79.3 | 70.2 | 73.9 | 70.6 | 75.1 | 74.1 | **75.9** |
| ATE | 68.7 | 54.7 | 62.1 | 68.2 | 55.7 | 70.3 | 63.0 | **71.6** |
| TE | 67.2 | 71.1 | 58.5 | 64.8 | 60.5 | 66.2 | 82.1 | **84.7** |
| PR | 73.5 | 73.7 | 69.5 | 52.1 | **80.5** | 79.9 | 69.0 | 65.3 |
| VN | 51.4 | 79.5 | 61.5 | 67.2 | 78.9 | 70.1 | 71.1 | **84.6** |
| BS | 64.7 | 67.2 | 58.5 | 65.5 | 77.1 | **80.7** | 70.2 | 79.7 |
| *mAP* (%) | 69.6 | 75.1 | 68.4 | 73.5 | 74.1 | 74.7 | 76.4 | **81.8** |

The following are the detection results of some typical samples on the LSDUVD dataset using SSD, FMSSD, Faster-RCNN, RetinaNet, YOLOv3, YOLOv4, YOLOv5s, and our method, as shown in Figures 13–16. In the detection task of drone aerial images, we select image types based on factors that affect the accuracy of object detection. For example, in Figure 13, we select image types where some targets are shaded. Due to differences in shadow area size and self-detection accuracy, the above models exhibit different results. Among them, the two-stage network Faster-RCNN, which lacks effective feature extraction, performs the worst, while our method YOLOv5s-Improved has the best detection rate and regression effect due to the presence of Gabor filter banks that enhance target edge characteristics.

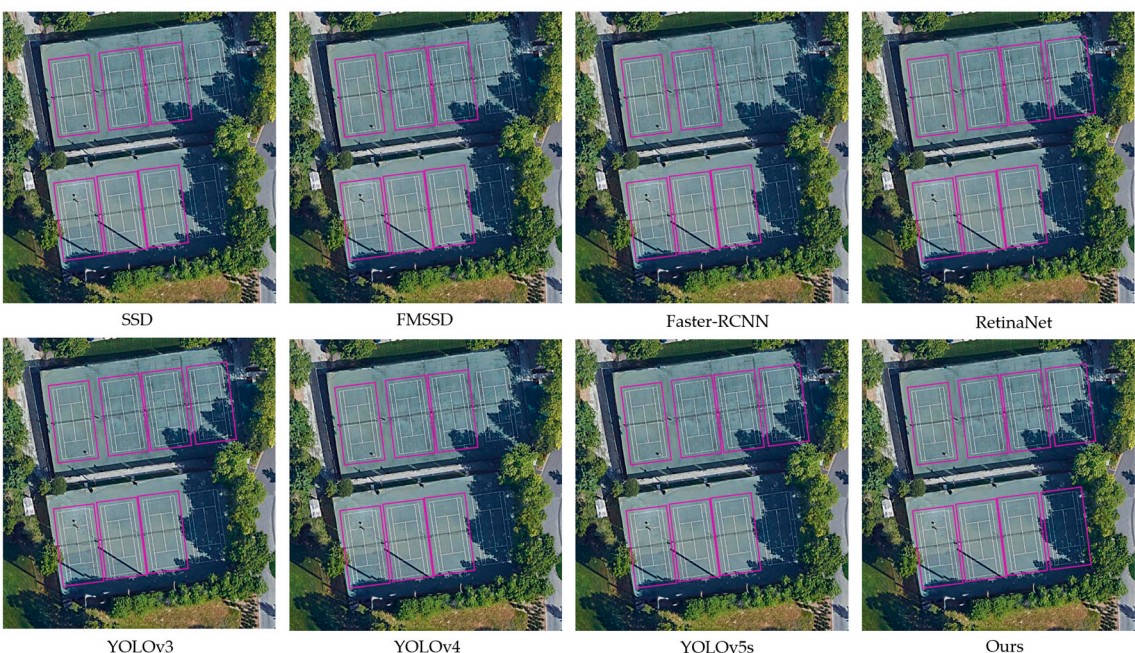

**Figure 13.** Visualization of detection results for image types where some objects are obscured by shadows.

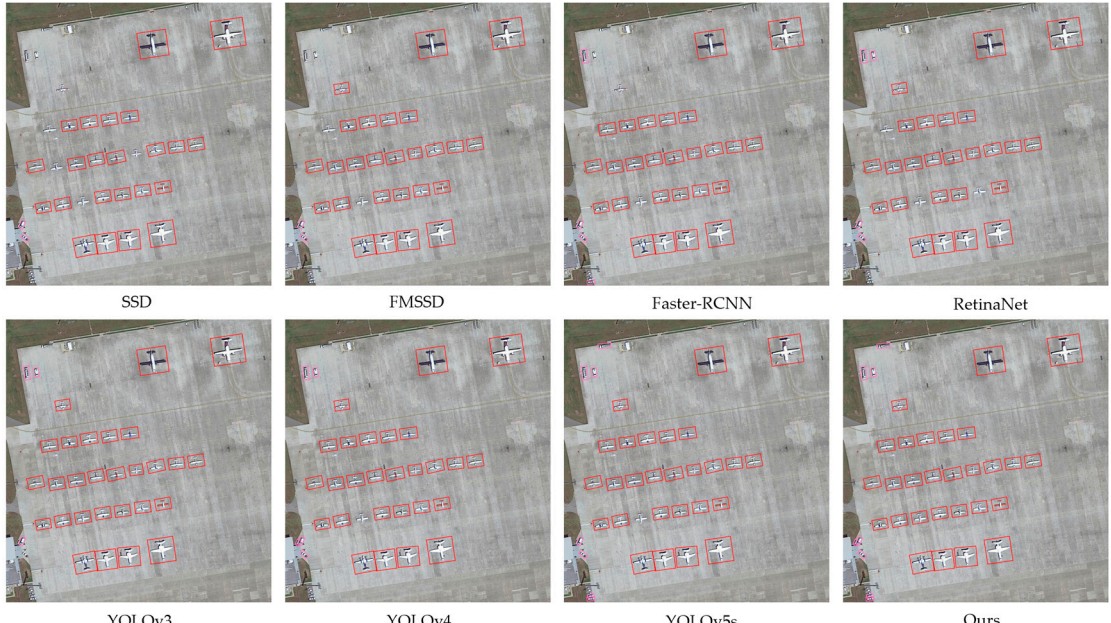

**Figure 14.** Visualization of detection results for image types with similar contrast between the object and background.

In Figure 14, we selected an image type with similar contrast between target and background. Due to the similarity in contrast between target and background, the pixel difference between target and background is not significant, and there is no clear boundary between the two. This makes it difficult for many models to effectively determine the position of the target due to their insufficient ability to filter background information and perceive the region. This leads to high missed detection rates for targets in this type of image in many networks. The edge reinforcement module and attention mechanism in YOLOv5s-Improved enable the network to effectively capture the feature information of

the target area, thus effectively solving such problems. As can be seen from Figure 14, our method has the best detection results compared with other methods.

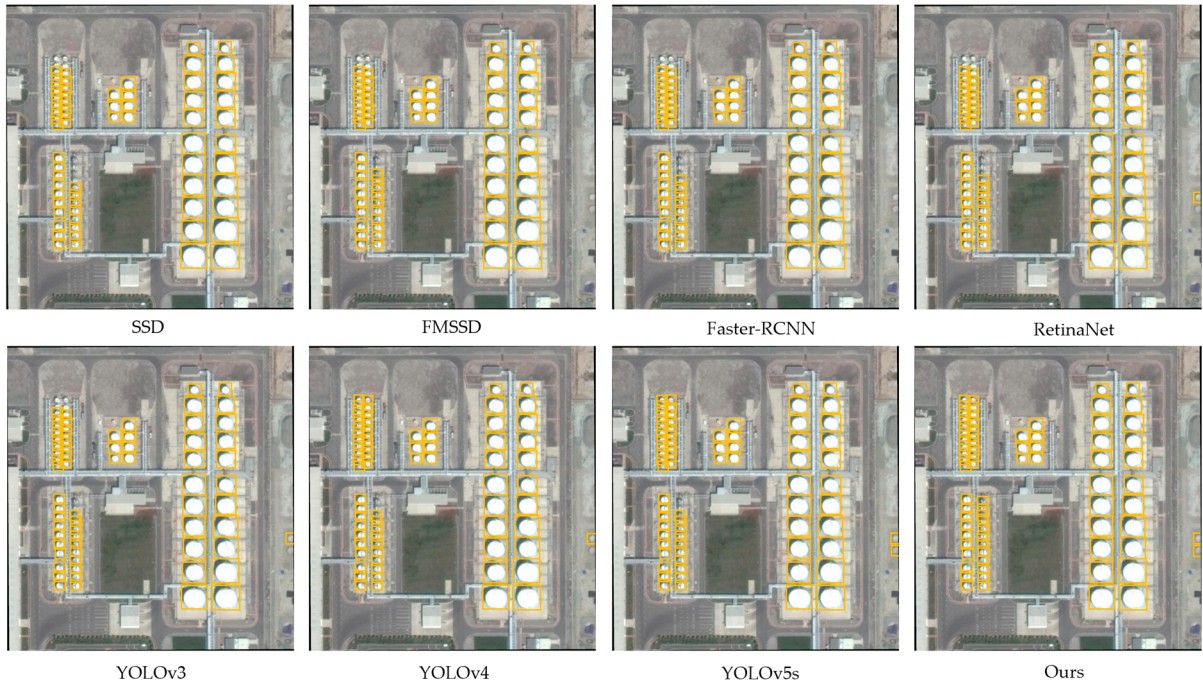

**Figure 15.** Visualization of detection results for densely arranged small objects and blurred background image types.

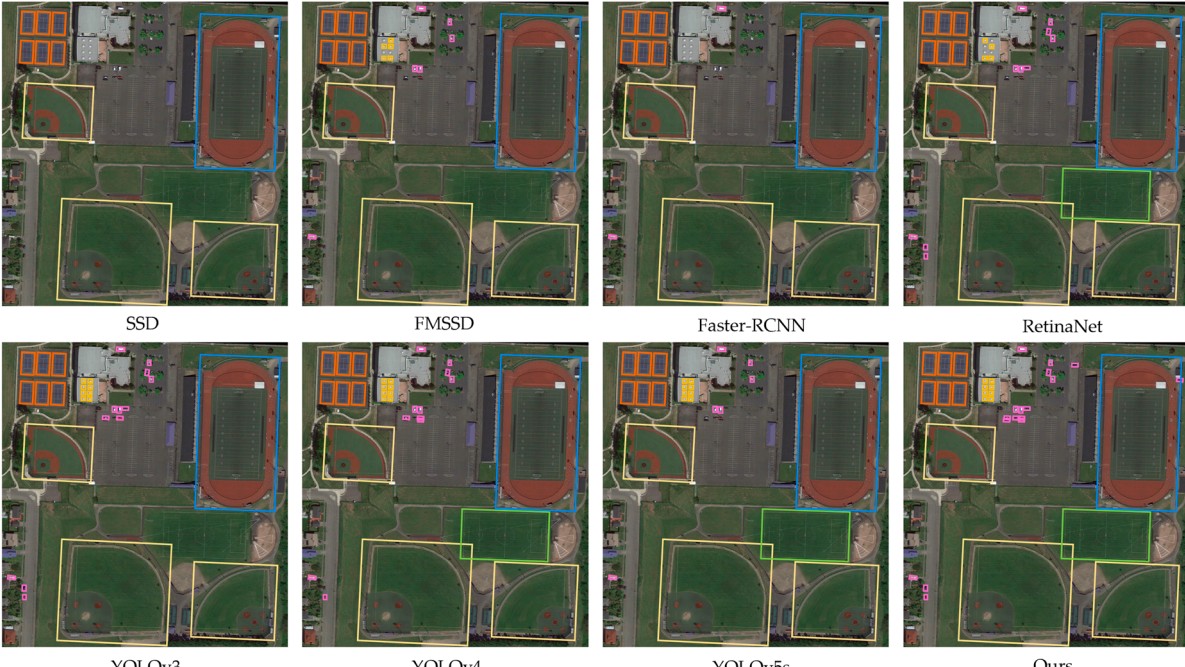

**Figure 16.** Visualization of detection results for multi-scale object image types.

In Figure 15, we selected an image type with densely arranged small objects and blurred backgrounds. From a subjective perspective, our method has the lowest missed detection rate, while other methods have varying degrees of missed detection. Some models, such as SSD, Faster-RCNN, and RetinaNet, have more severe issues with missed detection. From this, it can be inferred that densely arranged small targets and fuzzy backgrounds pose

a significant challenge to the model's detection ability, which requires the model to have a strong ability to extract small target features against fuzzy backgrounds. For our model, in the process of object edge enhancement through multi-directional Gabor convolution, not only does the target contour become clear but blurry background information is also filtered out. At the same time, combining CA and BiFPN mechanisms, the model can effectively extract and fuse multi-dimensional information for small object regions, minimizing the problem of sparse and difficult-to-detect small object pixels.

In Figure 16, we selected the image type of multi-scale objects. Overall, our model pays comprehensive attention to various levels and types of objects in multiple-scale object detection tasks, with not only the lowest missed detection rate but also the best regression effect between the predicted box and the real box. During the detection process, there are several types of targets: large-scale soccer ball fields with contour areas that are very similar to the background, small and densely arranged storage tanks, and small vehicles. These targets receive low attention from the model and are often overlooked during the detection process, which is the biggest difficulty for the model in multi-scale detection tasks. Our model, under the combined action of the CA mechanism and other modules, can fully integrate the feature information of various targets while considering obvious regional targets, paying more attention to targets with complex backgrounds and smaller scales. Compared to other models, our model can effectively overcome this problem.

As shown in Figure 17, in order to further demonstrate the comprehensive performance of our method, we selected aerial photographs of unmanned aerial vehicles with multiple scales and types of targets in large scenes for detection, and some detection results were enlarged and are displayed. Observing the detection results in the graph, it can be concluded that our method can not only detect cars and ships with significant scale changes but also has good detection performance for small cars that are shaded and densely arranged, or medium to large cars and ships that are similar to the background. It can be seen that our model has strong detection performance in processing complex types of drone aerial images.

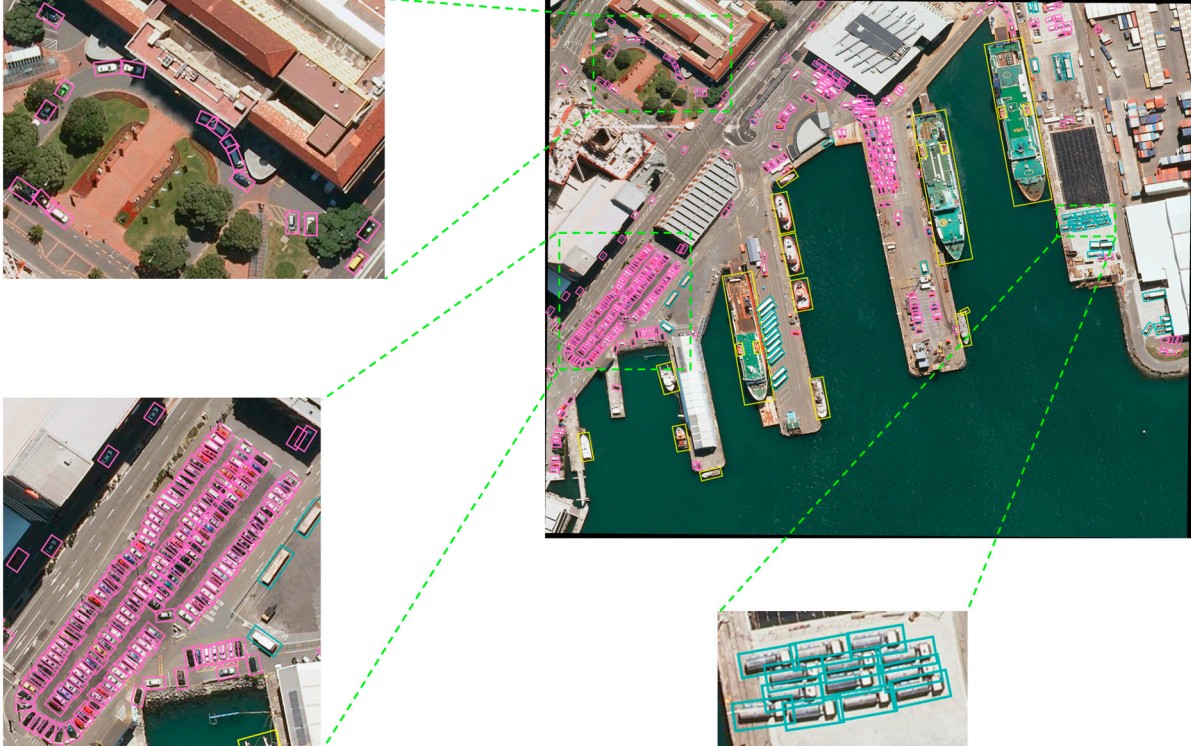

**Figure 17.** Visualization of the detection results at multiple scales and object types in large scenes in drone aerial images using our method.

## 5. Conclusions

This study was conducted to address the shortcomings of current drone aerial image object detection algorithms, aiming to solve the difficulties caused by complex backgrounds, object scale, and distribution issues in drone aerial images. We improved the YOLOv5 algorithm to boost the performance of object detection in drone aerial images. The main improvements were as follows. Firstly, we optimized the traditional Gabor function to obtain Gabor convolution kernels with better edge enhancement characteristics and use multiple Gabor convolution kernels to enhance the target edges from multiple directions, resulting in an enhanced image that was beneficial for subsequent deep feature extraction work. Secondly, we added a coordinate attention (CA) mechanism to the backbone of YOLOv5. This attention mechanism had the characteristics of being plug-and-play and lightweight, making it convenient for the operation of the network. At the same time, CA comprehensively considers the location information and channel information of features to accurately capture the long-range dependency relationship of positions, making it easier for the network to find the region of interest (ROI). Thirdly, we replaced the Path Aggregation Network (PANet) with the Bidirectional Feature Pyramid Network (BiFPN) on the neck of YOLOv5. BiFPN performed corresponding weighting operations based on the different contributions of each input layer in order to enrich the information of different feature layers. In addition, BiFPN added horizontally connected feature branches across nodes on a bidirectional feature fusion structure to fuse more and deeper feature information. Finally, we trained the YOLOv5-Improved model on our integrated dataset, LSDUVD, and compared it with other models on multiple datasets. The results showed that our method had efficient performance in processing detection tasks of drone aerial images.

In the future, we will continue to explore the characteristics of drone aerial images and pay attention to the difficulties encountered in object detection in drone aerial images. We will propose more targeted and innovative ideas to improve the performance of object detection algorithm models. In addition, continuously improving the category and quantity of test samples is also our key task.

**Author Contributions:** Conceptualization, H.Z., F.S. and X.H.; methodology and software, H.Z. and F.S.; validation and formal analysis, H.Z., Z.Z. and Y.C.; resources and data curation, X.H. and S.B.; writing—original draft preparation, review, and editing, H.Z., F.S. and X.H.; project administration, S.B.; funding acquisition, F.S. All authors have read and agreed to the published version of the manuscript.

**Funding:** This research was funded by the National Natural Science Foundation of China (grant number: 61671470).

**Data Availability Statement:** Not applicable.

**Conflicts of Interest:** The authors declare no conflict of interest.

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
