# Peer review of "Research on Object Detection and Recognition Method for UAV Aerial Images Based on Improved YOLOv5"

_drones, doi:10.3390/drones7060402_

Round 1

Reviewer 1 Report

Comments and Suggestions for Authors

Dear Authors,

I find the paper interesting. The proposed method is innovative, employing several new techniques such as improved Gabor function, coordinate attention mechanism and bidirectional feature pyramid network. The experiments are properly arranged, and the results are clearly analyzed, demonstrating the rationality and effectiveness of the proposed method. Overall, the work solves the problems of complex background, object scale, and object distribution of UAV aerial images, providing a method for object detection in UAV aerial images.

I would recommend the following revisions before publication:

1. I think the language of the manuscript needs further polishing. The vocabulary in the paper should be accurate, the structure of the sentence should be clear and scientific, and the grammar should be consistent.

For example:

1) The title uses the term "object detection", but the text uses the term "target detection" in several places;

2) Line10: “with complex backgrounds and object scale and distribution issues”, it may cause misunderstanding to readers;

3) Line 77: “we start by mainly solving the background and target distribution problems of drone aerial images are mainly solved regarding two aspects”, the grammar is inconsistent.

4) Line 501: “By investing greater weight in the target extraction area, highlighting and suppressing useful and useless features, respectively”, it may cause misunderstanding to readers;

And so on.

2. I think the tenses used in the vocabulary throughout the text should be reasonably modified. The authors should check it carefully.

3. In Section 3.3, the authors propose an improved YOLOv5 model and show the structure of improved YOLOv5s in Figure 3. However, the authors mentioned above that there are multiple versions of YOLOv5, such as YOLOv5s, YOLOv5m, YOLOv5l, and YOLOv5x, and I think the authors should explain the reason for choosing YOLOv5s.

4. In Figure 3, the authors show the structure diagram of the improved YOLOv5s, where I think the names of the input preprocessing flows should be labeled.

5. I think the authors should give a brief description of the training experimental parameters in Table 2.

6. Checking thoroughly for typos.

For example:

1) Line 284: Tt -- It;

And so on.

Comments on the Quality of English Language

Need further improvement, see comments.

Reviewer 2 Report

Comments and Suggestions for Authors

The paper addresses the important problem of object detection from images acquired by an unmanned aerial vehicle. To this end the authors propose an image pre-processing through a bank of Gabor filter and an enhanced version of YoloV5.

The paper includes all the needed details, from the related work up to the technical implementations and the experimental validation. Publicly sharing the code would be a valuable plus.

The Related works section is appreciably complete and detailed. However I would suggest to divide it into subsections addressing the different aspects of the proposed method. Furthermore I would suggest to include the following works:

https://link.springer.com/chapter/10.1007/978-3-319-68560-1_12

where the authors combined a classical computer vision method (the Hough transform) to improve a deep learning based model, openTLD for the task of autonomous landing.

In this other work the authors proposed vanishing point combined with YoloV2: https://ieeexplore.ieee.org/abstract/document/8451144  

In the following work PCA is used to pre-process the image for cars detection in aerial images through ResNet50 and MobileNetv1:  

https://link.springer.com/chapter/10.1007/978-3-031-27499-2_12

In Sec. 3.3 at line 439 the authors introduce nl. If my understanding is right it is the size of the discretized kernel of the Gabor filter. Probably explicitly mention this could improve readability.

Still in this section at line 454 the authors mention a two-scale and four-direction filter bank. It is unclear to me what the authors mean by two-scale. Probably this is also related to the rather unclear explanation in lines 447 to 450.

An important concern is related to the convolution kernel size mentioned in lines 455-457. How can the authors state to choose a single size for the kernel (5x5) with the large variety of object scales they have in their dataset (as shown in the experimental results)?

How is the simultaneous convolution mentioned in line 466 performed?

Lines 735-737: "Of course, this is not a definitive conclusion, but the experimental results of this article once again confirm our conclusion". This sentence seems contradictory. What do the author mean by "our conclusion" (also in line 795)? Moreover I would suggest to revise the entire sentence in lines 737-740 as it is unclear to me.

I was completely lost in the explanation in lines 801-807. What do the authors mean by "human subjective cognition"? Which "peripheral adjustments" do they mean?

Another important concern on this paper is related to the results shown in Table 4. As the authors state in lines 831-832, their model performs slightly poorly with small targets. However in the following images and discussion they underline how their method is capable of equally detecting both large and small targets in the same image (particularly in lines 888-892 and in Fig.s 16-17). A careful clarification on this aspet is needed.

Lines 860-861: "Among the many models mentioned above, our model has more advantages." Which advantages?

Minors:

Please introduce the full name for the most important acronyms the first time they are used (SSD is defined only in line 290 while it has been used several times in the lines above). I would suggest to introduce an Acronym section if possible.

Please recheck lines 77-78 as the sentence seems wrong.

Line 259: "based on their own features". Which features?

Line 261: "on the basis of the original method" . Which method?

Line 284: Tt should be "It"

Line 293: AAF should be "AFF"

Lines 418-419: The references to the formulas are exchanged.

Line 441 and Eq (4): please fix Lnegtive

Line 868: issued should be "issues"

Comments on the Quality of English Language

The paper is generally easy to follow. However some typos are present so I would suggest to have a careful reading. A proofreading of the English is suggested as some sentences are hard to understand or wrong. Some examples:

Line 79: the image preprocesses itself

Line 131 (and others): unmanned aerial vehicle aerial images

Reviewer 3 Report

Comments and Suggestions for Authors

Our comments are in the file attached.

Comments on the Quality of English Language

Round 2

Reviewer 1 Report

Comments and Suggestions for Authors

Dear Authors, 

  thanks for your revision, I have no further comments.

Reviewer 2 Report

Comments and Suggestions for Authors

The authors addressed nearly all my points. Thank you. Please find below two further comments:

Point 2:
In Sec. 3.3 at line 439 the authors introduce nl. If my understanding is right it is the size of the discretized kernel of the Gabor filter. Probably explicitly mention this could improve readability.

Response 2:  nl is an important parameter for discretization of convolution kernels, which can be understood as the dimension of discrete points. The purpose is to overcome the computational workload of traditional convolution kernels and improve the computational efficiency without obvious distortion in computational performance.

Thank you but please include this explanation in the manuscript.

Point 3: Still in this section at line 454 the authors mention a two-scale and four-direction filter bank. It is unclear to me what the authors mean by two-scale. Probably this is also related to the rather unclear explanation in lines 447 to 450.

An important concern is related to the convolution kernel size mentioned in lines 455-457. How can the authors state to choose a single size for the kernel (5x5) with the large variety of object scales they have in their dataset (as shown in the experimental results)?

Response 3: As for the two-scale and four-direction mentioned in the article, as shown in Figure 4, the two-scale here corresponds to the frequency or wavelength of Gabor function, which determines the width of Gabor convolution kernel texture from the visual point of view. The scale here is not the same object as the 5x5 we mentioned. 5x5 represents the 5x5 matrix of the selection center of the discrete Gabor kernel. We choose the 5x5 matrix because the values outside this region have little effect on the convolution results, but the filtering characteristics of the values within this matrix are still affected by the "two-scale and four-direction" of Gabor function.

Thank you but please include this explanation in the manuscript.
